# BoolNet: Streamlining Binary Neural Networks Using Binary Feature Maps

## Abstract

Recent works on Binary Neural Networks (BNNs) have made promising progress in narrowing the accuracy gap of BNNs to their 32-bit counterparts, often based on specialized model designs using additional 32-bit components. Furthermore, most previous BNNs use 32-bit values for feature maps and residual shortcuts, which helps to maintain the accuracy, but is not friendly to hardware accelerators with limited memory, energy, and computing resources. Thus, we raise the following question: *"How can accuracy and energy consumption be balanced in a BNN design?"* We extensively study this fundamental problem in this work and propose *BoolNet*: an architecture without most commonly used 32-bit components that uses 1-bit values to store feature maps. Experimental results demonstrate that *BoolNet* can achieve 63.0% Top-1 accuracy on ImageNet coupled with an energy reduction of $2.95\times$ compared to recent state-of-the-art BNN architectures. Code and trained models are available at: `(URL in final version)`

## 1 Introduction

The recent success of *Deep Neural Networks* (DNNs) is like the jewel in the crown of modern AI waves. However, the large size and the high number of operations cause the current DNNs to heavily rely on high-performance computing hardware, such as GPU and TPU. Training sophisticated DNN models also results in excessive energy consumption and $CO_2$ emission, e.g., training the OpenAI's GPT-3 by Brown et al. (2020) causes as much $CO_2$ emissions as 43 cars during their lifetime (Patterson et al., 2021). Moreover, their computational expensiveness strongly limits their applicability on resource-constrained devices such as mobile phones, IoT devices, and embedded devices. Various works aim to solve this challenge by reducing memory footprints and accelerating inference. We can roughly categorize these works into the following directions: network pruning (Han et al., 2015a;b), knowledge distillation (Crowley et al., 2018; Polino et al., 2018), compact networks (Howard et al., 2017; 2019; Sandler et al., 2018; Ma et al., 2018b; Tan et al., 2019), and low-bit quantization (Courbariaux et al., 2015; Rastegari et al., 2016; Zhou et al., 2016; Hubara et al., 2016). From the latter, there is an extreme case, Binary Neural Networks (BNNs), first introduced by Courbariaux et al. (2016), that uses only 1 bit for weight and activation.

As shown in the literature (Rastegari et al., 2016), BNNs can achieve $32\times$ memory compression and up to $58\times$ speedup on CPU, since the conventional arithmetic operations can be replaced by bit-wise `xnor` and `bitcount` operations. However, BNNs suffer from accuracy degradation compared to their 32-bit counterparts. For instance, XNOR-Net leaves an 18% accuracy gap to ResNet-18 on ImageNet classification (Rastegari et al., 2016). Therefore, recent efforts (analyzed in more detail in Section 2) mainly focus on narrowing the accuracy gap, including specific architecture design (Liu et al., 2018; Bethge et al., 2019; 2020; Liu et al., 2020b), real-valued weight and activation approximation (Lin et al., 2017a; Zhuang et al., 2019), specific training recipes (Martinez et al., 2020), a dedicated optimizer (Helwegen et al., 2019), leveraging neural architecture search (Bulat et al., 2020; Zhao et al., 2020) and dynamic networks (Bulat et al., 2021). In the existing work, efficiency analysis usually only considers the theoretical instruction counts. However, memory usage, inference efficiency and energy consumption, which are essential to practical applications, have received little attention. Furthermore, Fromm et al. (2020); Bannink et al. (2021) point out that the theoretical complexity is often inconsistent with the actual performance in practice and measurable performance gains on existing BNN models are hard to achieve as the 32-bit components in BNNs (such as BatchNorm, scaling, and 32-bit branches) become bottlenecks. Using 32-bit information flow (e.g.,

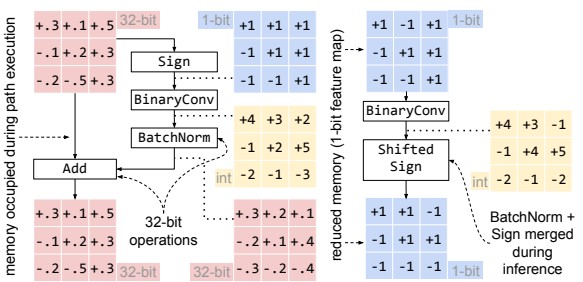

(a) Design in previous work.  (b) BoolNet design.

| Method | Bitwidth (W/A/F) | Energy (mJ) | Top-1 Acc. | OPs ($\cdot 10^8$) |
|---|---|---|---|---|
| Bi-Real-Net | 1/1/32 | 3.90 | 56.4% | 1.63 |
| BoolNet (ours) | 1/1/4 | 1.33 | 63.0% | 1.81 |
| BaseNet (ours) | 1/1/4 | 0.83 | 58.2% | 1.54 |
| BaseNet (ours) | 1/1/1 | 0.70 | 53.3% | 1.51 |

(c) BoolNet reduces the energy consumption by $2.9\times$ compared to Bi-RealNet by Liu et al. (2018).

Figure 1: The main differences between previous work and BoolNet. BoolNet uses 1-bit feature maps and a shifted sign function reducing memory requirements and the need for 32-bit operations.

32-bit identity connections, 32-bit downsampling layers are equipped by almost all latest BNNs, see Figure 1a), and multiplication/division operations (in BatchNorm, scaling, average pooling etc.) significantly increase the memory usage and power consumption of BNNs and are thus unfriendly to hardware accelerators. For these reasons, even if BNNs have achieved MobileNet-level accuracy with a similar theoretical number of OPs (Bethge et al., 2020; Martinez et al., 2020), they still cannot be used as conveniently as compact networks (Howard et al., 2017; 2019; Sandler et al., 2018).

In this paper, we extensively study the trade-off between BNN's accuracy and hardware efficiency. We propose a novel BNN architecture: BoolNet, which replaces most commonly used 32-bit components (see Section 3). First, BoolNet only uses binary feature maps in the network (see Figure 1b). Second, during inference, we fuse the BN layer into the Sign function through a lossless transformation, thereby effectively removing the Mult-Adds brought by BN. Other changes include removing components that require additional 32-bit multiplication/division operations: (1) PReLU, (2) average pooling, and (3) binary downsampling convolutions. We then propose a *Multi-slice strategy* to help alleviate the loss of representational capacity incurred by binarizing the feature maps and removing 32-bit components. We show the effectiveness of our proposed methods and the increased energy efficiency of BoolNet with experiments on the ImageNet dataset (Deng et al., 2009). The results show the key benefit of BoolNet: a reasonable accuracy coupled with a higher energy efficiency over state-of-the-art BNNs (see Figure 1c for a brief summary and Section 4 for more details). The energy data is obtained through a hardware accelerator simulation (see Section 4.4 for details). We summarize our main contributions as follows:

- The first work studying the effects of 32-bit layers often used in previous works on BNNs.
- A novel BNN architecture BoolNet with minimal 32-bit components for higher efficiency.
- A Multi-slice strategy to alleviate the accuracy loss incurred by using 1-bit feature maps.
- State-of-the-art performance on the trade-off between accuracy and energy consumption with a $2.9\times$ lower power consumption than Bi-RealNet (Liu et al., 2018) and 6.6% higher accuracy.

## 2  RELATED WORK

In recent years, *Efficient Deep Learning* has become a research field that has attracted much attention. Technical directions, such as, compact network design (Howard et al., 2017; 2019; Sandler et al., 2018; Zhang et al., 2018; Ma et al., 2018b), knowledge distillation (Crowley et al., 2018; Polino et al., 2018), network pruning (Han et al., 2015a;b; Li et al., 2017; He et al., 2017), and low-bit quantization (Courbariaux et al., 2015; Rastegari et al., 2016; Liu et al., 2018; 2020b; Bethge et al., 2020) are proposed for model compression and acceleration. The efficient models have evolved from the earliest handcrafted designs to the current use of neural architecture search to search for the best basic block and overall network structure (Tan et al., 2019; Howard et al., 2019; Tan & Le, 2019; Radosavovic et al., 2020). The criterion of efficiency evaluation has also changed from instruction and parameter counts to more precise measurements of actual memory and operating efficiency on the target hardware (Cai et al., 2019; 2018).

Binary Neural Networks were first introduced by Courbariaux et al. (2016) and their initial attempt only evaluated on small datasets such as MNIST (LeCun & Cortes, 2010), CIFAR10 (Krizhevsky et al., 2009) and SVHN (Netzer et al., 2011). The follow-up XNOR-Net (Rastegari et al., 2016) proposes channel-wise scaling factors for approximating the real-valued parameters, which achieves 51.2% top-1 accuracy on ImageNet. However, there is an 18% gap compared with its 32-bit counterpart, ResNet-18. Therefore, recent efforts mainly focused on narrowing the accuracy gap. WRPN by Mishra et al. (2018) shows that expanding the channel width of binary convolutions can obtain a better performance. ABC-Net by Lin et al. (2017a), GroupNet by Zhuang et al. (2019), and (Zhu et al., 2019) use a set of $k$ binary convolutions (referred to as binary bases), instead of using a single binary convolution, to approximate a 32-bit convolution. This sort of method achieves higher accuracy but increases the required memory and number of operations of each convolution by the factor k. Bi-RealNet by Liu et al. (2018) proposes using real-valued (32-bit) shortcuts to maintain a 32-bit information flow, which effectively improves the accuracy. This design strategy became a standard for later work, e.g., Bethge et al. (2019; 2020); Liu et al. (2020b). Martinez et al. (2020) propose using a real-valued attention mechanism and well-tuned training recipes to boost the accuracy further. Thanks to the special architecture design, the recent MeliusNet (Bethge et al., 2020) and ReActNet (Liu et al., 2020b) achieve MobileNet-level accuracy with similar number of theoretical operations. Other attempts, such as leveraging neural architecture search (Bulat et al., 2020; Zhao et al., 2020) and dynamic networks (Bulat et al., 2021), show that those successful methods on regular real-valued networks are also effective for BNN. Another method by Shen et al. (2019) combines neural architecture search to dynamically increase the number of channels for more accurate BNNs. Often, with improved accuracy, 32-bit components are used more frequently as well, such as PReLU and BatchNorm after each binary convolution (Liu et al., 2020b), a real-valued attention module (Martinez et al., 2020) and scaling factors, etc. Apart from some works that include and optimize real-time measurements on mobile devices, such as Bannink et al. (2021); Umuroglu et al. (2017), efficiency analysis in the literature often only considers the theoretical operation number. However, the memory usage and the actual energy consumption has received very little attention so far.

## 3 BOOLNET

In this section, we first revisit the latest BNNs and recap how they enhanced the accuracy by adding more 32-bit components (in Section 3.1). Afterwards, we propose to replace most commonly used 32-bit components from current BNN designs and instead use a fully binary information flow in the network (in Section 3.2). However, abandoning 32-bit information flow results in a serious degradation of the representative capacity of the network. Thus, we also present our strategies to restore the representative capacity (in Section 3.3). The focus on boolean operations and binary feature maps leads to the name of our network: **BoolNet**.

### 3.1 IMPROVING ACCURACY WITH ADDITIONAL 32-BIT COMPONENTS

Recent works on BNNs have made promising progress in narrowing the gap to their 32-bit counterparts. The key intention is to enhance the representative capacity by fully exploiting additional 32-bit components. However, such additional 32-bit components significantly reduce the hardware efficiency (as shown by Fromm et al. (2020) and further discussed in Section 4.4). The following list summarizes the 32-bit components commonly used in the latest BNNs: (1) The **channel-wise scaling factor** was first proposed by Rastegari et al. (2016) for approximating the 32-bit parameters. It increases the value range of activation and weight. (2) Bi-RealNet (Liu et al., 2018) proposes to use a **32-bit shortcut** for enclosing each binary convolution. The key advantage is that the network can maintain an almost completely 32-bit information flow (cf. Figure 2a). (3) XNOR-Net (Rastegari et al., 2016) uses **32-bit 1×1 downsampling** convolutions, which is also used by many subsequent methods (Liu et al., 2018; Martinez et al., 2020; Bethge et al., 2020). Bethge et al. (2019) shows that this simple strategy can achieve about 3.6% Top-1 accuracy gains on ImageNet based on a binary ResNet-18 model. (4) Martinez et al. (2020); Bulat et al. (2020; 2021) show that **PReLU activation** effectively improves accuracy of BNNs. ReActNet (Liu et al., 2020b) constructs the RPReLU activation function and uses it before every sign function. (5) Martinez et al. (2020) reuse the 32-bit activation in their Real-to-Binary Net after BN with a squeeze and excitation (SE) **attention mechanism**. This module can adaptively re-scale the outputs of each binary convolution but needs additional 32-bit operations.

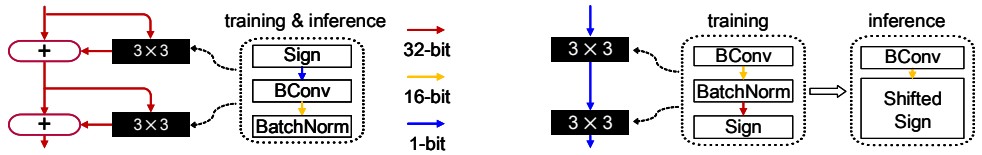

(a) Typical binary basic block.      (b) Our binary block design without 32-bit operations.

Figure 2: Comparison between a conventional binary convolution block with 32-bit data flow (a) and our proposed binary convolution block with 1-bit data flow (b).

Although these techniques can effectively improve the accuracy, they increase the number of 32-bit values and floating point operations, making them not particularly efficient on hardware accelerators. They are closer to mixed-precision neural networks rather than being highly efficient binary neural networks, as one might expect.

## 3.2 BASENET: REPLACING 32-BIT COMPONENTS WITH BOOLEAN OPERATIONS

To better balance accuracy and efficiency, we rethink the additional 32-bit components (Batch Normalization, 32-bit feature maps, scaling factors and PReLU) elaborated in the previous section and propose to remove or replace them with more efficient operations. We further propose a new basic convolution block without 32-bit operations, as shown in Figure 2b, where we rearranged the order of convolution basic block as [BinaryConv, BatchNorm, Sign], so that all feature maps are binary. These general changes constitute our BoolNet baseline, in short *BaseNet*.

**Integrating the BatchNorm into the Sign Function** Umuroglu et al. (2017) suggested to replace the BatchNorm (BN) with a thresholding operation during inference on FPGAs. However their suggestion can not be applied to more recent work (Hubara et al., 2016; Rastegari et al., 2016; Liu et al., 2018; 2020b; Bethge et al., 2020), because the layer order in these works is [Sign, BinaryConv, BN] surrounded by 32-bit valued shortcuts. Instead, these recent works have kept the 32-bit BatchNorm layer in both the training and testing stages. However, using a 32-bit BN right after the 1-bit convolution layer decreases the computational efficiency on hardware, using more memory and energy. Thus, in the following we propose to fuse the BN layer into the Sign function during the inference stage and do not use the 32-bit output of BN layer for shortcut connection.

During the training phase, the batch normalization layer normalizes feature maps with a running mean $\mu$ and a running variance $\sigma$. For inference, it utilizes the constant statistic mean and variance instead, which in result can be reformulated as a linear process, expressed as:

$$y_i = \gamma \frac{x_i - \mu}{\sqrt{||\sigma^2 + \epsilon||}} + \beta = \frac{\gamma}{\sqrt{||\sigma^2 + \epsilon||}} x_i + \left( \beta - \frac{\gamma\mu}{\sqrt{||\sigma^2 + \epsilon||}} \right) \tag{1}$$

where $x_i$ and $y_i$ represent the N-dimensional input and output of a BN layer. $\gamma$ and $\beta$ are trainable scale and shift parameters, which are constant during the inference. $||\dots||$ is the absolute function. We can therefore simplify the formula as follows:

$$y_i = ax_i + b \;=\; a\left(x_i + \frac{b}{a}\right) = a\left(x_i + c\right) \;, \tag{2}$$

where $a$, $b$, and $c$ denote constants in the formula. By transforming $a$ into its sign and its absolute value, we have

$$y_i = ||a|| \circledast \mathrm{Sign}\,(a) \odot (x_i + c), \tag{3}$$

As arranged in our basic block, Equation (3) is followed by a sign function, and $\mathrm{Sign}(y_i)$ only depends on $\mathrm{Sign}(a)$ and $(x_i + c)$. We thus derive a parameterized sign function as:

$$\mathrm{Sign}(y_i) = \mathrm{XNOR}(\mathrm{Sign}(a), \mathrm{Sign}(x_i + c)) \tag{4}$$

We further replace $\odot$ by using XNOR operator so that only bit-wise operations are adopted in the inference.

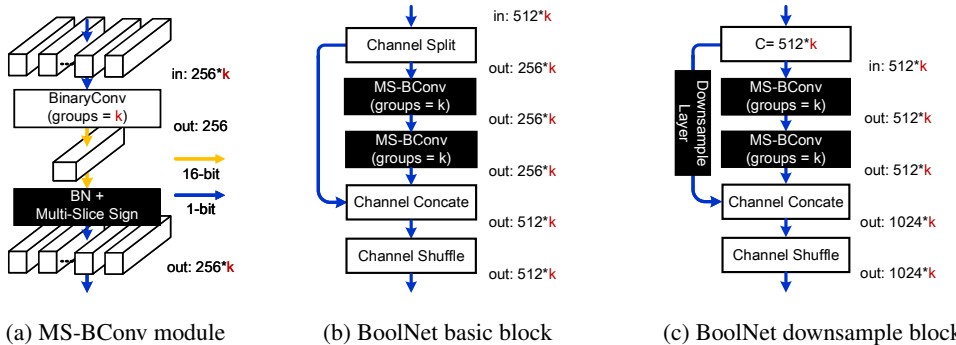

(a) MS-BConv module      (b) BoolNet basic block      (c) BoolNet downsample block

Figure 3: The detailed architecture of BoolNet. To enhance the information flow, we modify the baseline architecture in two aspects: a) Reducing information loss with our multi-slices binary convolution. b) Strengthening the information propagation by using split and concatenate operations.

**Further Reducing 32-bit Operations** We rarely use the PReLU activation function, which is commonly used in the recent literature (Liu et al., 2018; Martinez et al., 2020; Bulat et al., 2021) and brings a lot of extra overhead to the hardware implementation (it is only used once before the final dense layer). We also decided not to use scaling factors as suggested by Liu et al. (2018); Bethge et al. (2019). There are two components which use 32-bit operations and parameters in previous work, which are kept in 32-bit in BoolNet: the first convolution and the last dense layer. Directly replacing them with binary versions leads to a severe accuracy loss (Rastegari et al., 2016), thus we leave the investigation of alternatives for these special layers for future work.

### 3.3 BOOLNET: ENHANCING BINARY INFORMATION FLOW

The network design changes explained in the previous section, constitute our BoolNet baseline, called *BaseNet*. Although it uses a completely binary information flow which minimizes the energy and memory consumption, the representative capacity of BaseNet is drastically degraded compared to its 32-bit counterparts and previous BNN methods, which accumulate information in 32-bit values. To counter this reduction of representative capacity, we propose the following three ideas, which constitute our proposed **BoolNet**.

**Multi-slice Binary Convolution**. Instead of using a single 1-bit value for each 32-bit value in a regular BNN, our multi-slice strategy proposes of using a set of $k$ 1-bit values. The key intention is to reduce the information loss caused by the sign function. We consider the typical binarization process $\text{Sign}(x_i, \text{zero-point})$ as a special case of single-slice numerical projection. Thus, we propose a multi-slice projection strategy for binary convolution to retain more relative magnitude information. Specifically, we redesign sign function as follows:

$$x_i^b = \text{Sign}(x_i, b_n), \tag{5}$$

where $b_n$ indicates a set of constant bias:

$$b_n = \frac{\pm 2n}{k}, \text{where } n = 0, 1, ..., k/2 \tag{6}$$

We adopt $b_n$ to conveniently expand the channel dimension to enhance the capacity of the binary feature map. If $n = 0, k = 1$, Equation (6) degenerates to the ordinary sign function. In Equation (5), $x_i^b$ denotes the binary projection output with the dimension of $[N, C * k, H, W]$, which will be fed into the subsequent binary convolution layer. The constant $k$ also denotes the group number of the convolution. That is, by setting the number of groups to $k$ in each convolution, the overall amount of parameters and operations of each convolution is unchanged. We found three conceptually related works by Lin et al. (2017b); Zhuang et al. (2019); Zhu et al. (2019), but they have a different motivation and implementation compared to our work: 1) They use a higher number ($k$) of 32-bit feature maps *and* weights to close the accuracy gap to 32-bit networks. This increases the computation *and* memory cost of a single convolution by the factor $k$ ($k^2$ in (Lin et al., 2017b)). 2) These works use differing composition strategies, but they usually require additional 32-bit computation, such as

addition and 32-bit scaling factors. For comparison, the Multi-slice Binary Convolution *reduces* the memory requirement by replacing each 32-bit value with $k$ 1-bit values and uses a grouped binary convolution to keep the number of operations constant. Motivated by FReLU (Ma et al., 2020), we enhance the first multi-slices projecting module, after the input convolution of network, with a **Local Adaptive Shifting** module. This module consists of a binary depth-wise $3 \times 3$ convolution and a batch normalization layer and is able to adaptively change the zero points of each pixel, in a light-weight manner. For simplicity, the multi-slices binary convolution is referred to as **MS-BConv**, subsequently. Figure 3a shows the detailed block design of MS-BConv.

**Strengthening The Information Propagation in BoolNet**. The layer-by-layer feature extraction and accumulation mechanism are key reasons deep neural networks have strong representative capacity. However, accumulating information from shallow to deep based on addition operation involves extra 32-bit data-flow, which is conflicted with our motivation of building hardware friendly binary neural network. To avoid 32-bit data-flow while maintaining the accumulation mechanism, we strengthen *BaseNet* by reusing binary features. In ShuffleNet-V2 (Ma et al., 2018a), the input tensor is divided into two equal parts, the first half is used for feature extraction, and the other half is directly copied and concatenated with the extracted features. Inspired by its characteristics of information fusion and retention, we use a similar method to enhance the information retention capability of the BoolNet block. As demonstrated in Figure 3b, the feature extraction branch consists of two MS-BConv modules, and the other branch remains identity. Two branches are concatenated and followed by channel shuffle, ensuring that the features from different layers are uniformly distributed. Figure 3c shows the downsampling block design of BoolNet, where no channel splitting is required, and it doubles the number of channels in the output. Changing this information accumulation mechanism constitutes our proposed **BoolNet** over the *BaseNet* (as referred to in Section 4).

**Rethinking the Downsample Branch**. Furthermore, we modify the downsampling block compared to previous methods, which usually use the layers [2×2 AvgPool (Stride 2), 32-bit 1×1 Conv, BN] in this branch (Liu et al., 2018; Martinez et al., 2020). Instead, we propose to use the layers [1-bit 1×1 Conv, 2×2 MaxPool (Stride 2), BN, Sign] and replace the 1-bit 3×3 Conv (stride 2) in the main branch with [1-bit 3×3 Conv, 2×2 MaxPool (stride 2)]. Overall, this removes all 32-bit operations and 32-bit parameters from the downsample block of BoolNet, but due to space limitations, we discuss the details in the appendix (including alternatives and experimental results).

### 3.4 Training with Progressive Weight Binarization

Though we intend to build highly efficient BNNs with fully binary information flow, this strategy makes the network more sensitive to weight initialization during training. Traditional methods tried to alleviate similar problems through two-stage training (Martinez et al., 2020; Liu et al., 2020b), which makes the training more complicated. In this paper, we adopt a progressive binarization technique based on the traditional Hardtanh-STE method (Courbariaux et al., 2016). This can be understood as a smooth version of a multi-stage training approach. Specifically, in the training phase, a differentiable function $F(x)$ is used to replace sign function. The slope of this function is adjusted by a single scalar $\lambda$. As the slope increases, the weight gradually changes from 32-bit to 1-bit. During backward propagation, we approximate $F(x/\lambda)$ with $F(x/1)$, to avoid the problem of gradients clipping as $\lambda$ decreases. In the testing phase, we use the regular sign function for inference. The whole process can be formulated as:

$$F(x, \lambda) = \lim_{\lambda \to 0} \text{Hardtanh}\left(\frac{x}{\lambda}\right) \simeq \text{Sign}(x). \tag{7}$$

To smooth the weight binarization process, we schedule $\lambda$ during training with an exponential decay strategy $\lambda_t = \sigma^{(t)}$, where $\sigma < 1$ is the exponential decay rate of $\lambda$.

## 4 Experiments

We evaluated our baseline network *BaseNet* and our proposed *BoolNet* (as described in Sections 3.2 and 3.3, respectively) on the ImageNet dataset (Deng et al., 2009). In the following section, we first present the training details for our experiments. Afterwards, we conduct an ablation study on our proposed network design changes and the *Multi-slice* convolution (in Section 4.2 and 4.3), analyze the energy consumption of BoolNet and other recent work on BNNs (in Section 4.4), and compare our model accuracy to state-of-the-art BNN models (in Section 4.5).

Table 1: Our ablation study on ImageNet (Deng et al. (2009)) regarding accuracy, number of 32-bit operations (FLOPs), 1-bit operations (BOPs), and model size. (OPs = FLOPs + $1/64$·BOPs)

| Network Configuration | BaseNet | | | | | BoolNet | | | | |
|---|---|---|---|---|---|---|---|---|---|---|
| | Top 1 Acc. | FLOPs ($\cdot10^8$) | BOPs ($\cdot10^9$) | OPs ($\cdot10^8$) | Model Size | Top 1 Acc. | FLOPs ($\cdot10^8$) | BOPs ($\cdot10^9$) | OPs ($\cdot10^8$) | Model Size |
| Baseline (60 epochs, CE Loss) | 47.69% | 1.22 | 1.68 | 1.49 | 3.47 MB | 54.07% | 2.78 | 1.85 | 3.07 | 5.05 MB |
| + Multi-Slice strategy (k=4) | 52.27% | 1.22 | 1.69 | 1.49 | 3.47 MB | 56.84% | 2.78 | 1.86 | 3.07 | 5.05 MB |
| + (1) Modified downsample branch | (BaseNet has no downsample branch) | | | | | 58.66% | 1.23 | 2.48 | 1.62 | 3.84 MB |
| + (2) Local Adaptive Shifting† | 52.08% | 1.25 | 1.69 | 1.51 | 3.47 MB | 59.56% | 1.26 | 2.48 | 1.65 | 3.84 MB |
| + (3) MaxPool instead of stride | 55.14% | 1.23 | 2.21 | 1.57 | 3.47 MB | 59.98% | 1.26 | 3.53 | 1.81 | 3.84 MB |
| + (4) Knowledge distillation‡ | 56.84% | 1.23 | 2.21 | 1.57 | 3.47 MB | 61.98% | 1.26 | 3.53 | 1.81 | 3.84 MB |
| + Long training (256 epochs) | 58.20% | 1.23 | 2.21 | 1.57 | 3.47 MB | 63.00% | 1.26 | 3.53 | 1.81 | 3.84 MB |

† Local Adaptive Shifting is not used for the subsequent BaseNet experiments    ‡ Replaces the cross-entropy loss with the distributional loss by Liu et al. (2020b)

| k | BaseNet baseline + (3) + (4) | | | | | | | BoolNet baseline + (1) + (2) + (3) + (4) | | | | | | |
|---|---|---|---|---|---|---|---|---|---|---|---|---|---|---|
| | Top 1 Acc. | Δ | Top 5 Acc. | FLOPs ($\cdot10^8$) | BOPs ($\cdot10^9$) | OPs ($\cdot10^8$) | Model Size | Top 1 Acc. | Δ | Top 5 Acc. | FLOPs ($\cdot10^8$) | BOPs ($\cdot10^9$) | OPs ($\cdot10^8$) | Model Size |
| 1 | 51.74% | - | 75.39% | 1.23 | 2.20 | 1.57 | 3.47 MB | 57.62% | - | 80.47% | 1.26 | 3.05 | 1.74 | 3.71 MB |
| 2 | 55.75% | +4.01 | 79.08% | 1.23 | 2.20 | 1.57 | 3.47 MB | 60.57% | +3.95 | 82.56% | 1.26 | 3.21 | 1.76 | 3.76 MB |
| 4 | 56.84% | +1.09 | 79.85% | 1.23 | 2.21 | 1.57 | 3.47 MB | 61.98% | +1.41 | 83.75% | 1.26 | 3.53 | 1.81 | 3.84 MB |
| 8 | 57.19% | +0.35 | 80.33% | 1.23 | 2.22 | 1.58 | 3.47 MB | 62.54% | +0.56 | 84.14% | 1.26 | 4.16 | 1.91 | 4.01 MB |

## 4.1 Training Details

Our training strategy and hyperparameters are mostly based on Bethge et al. (2020), but we train for 256 epochs and replace the Cross-Entropy loss with the knowledge distillation approach proposed by Liu et al. (2020b) where the teacher model is a 32-bit ResNet-34 (He et al., 2016). The exact hyperparameters, details and code are available in the appendix and the supplementary material.

We proposed progressive weight binarization (see Section 3.4, Equation 7) as an alternative to the two-stage training approach taken by Liu et al. (2020b); Martinez et al. (2020). For our experiments, we use $\sigma = 0.965$ and thus $\lambda = 0.965^t$, where $t$ is the number of samples processed divided by 256000, rounded down. The progressive weight binarization is replaced with the sign function during the validation pass. The two-stage training strategy aims to provide a good initialization for a BNN training, by first training a model with 1-bit activations/32-bit weights and weight decay of $10^{-5}$, and use it to initialize the training of a 1-bit activations/1-bit weights model. We tested the effect of both strategies with a plain ResNet-like model with binary feature maps. The two-stage training (trained 60 epochs in each stage - a total of 120 epochs) achieved 49.60% accuracy. Our progressive weight binarization achieves 48.39% when training for **60** epochs, but achieves 50.19% when training for **120** epochs. Thus we deduce that our training strategy effectively removes the need for a two stage training (based on a similar total training time) and leads to a similar or slightly better result.

## 4.2 Ablation on Network Design and Training Strategies

We conducted an ablation study to determine the influence of our proposed design changes and the used training techniques (see the top half of Table 1). Our results show a 4.58% (BaseNet) / 2.77% (BoolNet) increase of accuracy by using our Multi-Slice strategy with $k = 4$ over our baseline training (both trained only for 60 epochs with cross-entropy loss). Although adopting our *BoolNet over BaseNet* increases accuracy by 4.57% / 6.38% (with/without the Multi-Slice strategy), BoolNet uses a much higher number of 32-bit operations. However, *modifying the downsample branch* (as proposed in Section 3.3) eliminates these additional 32-bit operations in BoolNet and boosts accuracy by 1.82%. The *Local Adaptive Shifting* module increases accuracy by 0.90% for BoolNet but does not increase the accuracy of a BaseNet (and thus was not used for our subsequent BaseNet experiments). The reason is likely that the module only shifts the zero points of the very first block of BaseNet which does not aid optimization. Using a *MaxPool* layer (see Section 3.3) instead of a strided convolution especially helps BaseNet to retain information (which has no traditional shortcut branch in the downsampling block).

Regarding the training techniques, we found that replacing the cross-entropy loss with the knowledge distillation approach by Liu et al. (2020b) contributes 1.70% (BaseNet) / 2.00% (BoolNet) accuracy and extending the training time from 60 to 256 epochs further increases accuracy by 1.36% (BaseNet)

/ 1.02% (BoolNet). We conclude that all proposed techniques (and the knowledge distillation by Liu et al. (2020b)) help to increase the accuracy of BaseNet and/or BoolNet.

## 4.3 ABLATION ON THE MULTI-SLICE BINARY CONVOLUTION

The previous section has already shown that our Multi-slice Binary Convolution (see Section 3.3) can reduce the accuracy loss caused by using 1-bit feature maps. However, we further evaluated the influence of the number of slices $k$ in these convolutions based on the best configuration of the previous section but only training for 60 epochs (see the lower half of Table 1). Our results show that the final accuracy increases with each increase of $k$, but there are diminishing returns. Doubling $k$ from 1 to 2, from 2 to 4, and from 4 to 8 increases accuracy by 3.95%, 1.41%, and 0.56% respectively (for BoolNet). Based on the increase of the number of operations, model size, and projected memory consumption, we use $k = 4$ for the best trade-off between accuracy and energy efficiency for our following experiments.

## 4.4 ENERGY CONSUMPTION EVALUATION

This section aims to efficiently compare the energy consumption of BoolNet to classic BNN architectures under strictly fair design conditions. We thus implemented five BNN accelerators (BaseNet, BoolNet, XNOR-Net, Bi-RealNet, ReActNet (based on a Bi-RealNet backbone)). Considering the scope of this work, we leave the details of further hardware optimization of individual accelerators for future work.

We designed the five accelerators in the RTL language. Then, the power and area of computing circuits is given by Design Compiler (DC) with a TSMC 65nm process and 1GHz clock frequency. We refer to the design and implementation methods of computing units of Conti et al. (2018); Zhang et al. (2021). For a fair comparison between the different BNNs, we keep the design of architecture, data stream, the parallelism of computing units, and total on-chip cache (192KB for feature maps and 288KB for weights) consistent and only change the bit-width of the data stream and computing units. More specifically, the parallelism of binary convolution is $64 \times 64$, and the parallelism of other units is 64 in all accelerators (except the IntConv module is $8 \times 64$). These modules are pipelined and run at 1GHz. When DRAM bandwidth can be fully utilized, the performance depends on the parallelism and is bounded by the convolution, so each accelerator has the same peak performance for convolution, i.e., 4096 GOPs/s. Therefore, we achieve similar throughputs between 2044 and 2237 samples per second and it is reasonable to compare the energy consumption of the whole inference process.

DC can provide the hierarchical circuit area and power of computing units, including static power ($P_s$) and dynamic power ($P_d$). For each layer of the network, we know the amount (A) of each operation. According to the circuit parallelism ($P_a$), we can calculate the required number of cycles ($C_n$ = A / $P_a$) and then the energy consumption according to the frequency and power ($E_c = C_n \times (P_s + P_d) / 10^{-9}$). For the operations with fewer cycles, the energy consumption waiting for other units is estimated by static power ($E_s = (C_n^{max} - C_n) \times P_s / 10^{-9}$). Similar to Wang et al. (2021); Jiang & Zokaee (2021); Li et al. (2021), we evaluate the energy consumption of on-chip SRAM access and off-chip DRAM access by using CACTI 6.5 (CACTI) and the power calculator of DDR provided by Micron (Micron). According to the cycle accurate simulation above, we can get the access amount of SRAM and DRAM, then get the energy consumption. The above components sum the overall energy consumed by a single inference pass. More details of simulation and energy estimation can be found in A.3.

Memory access and computation are the primary factors that affect energy consumption of a hardware accelerator. In the existing BNNs, efficiency analysis only considers the theoretical instruction counts, while the impact of memory access is often not considered. However, Yang et al. (2017) has shown that computation energy only accounts for 10% of the overall energy consumption of a (32-bit) GoogleNet. Moreover, Han et al. (2015b) evaluated the energy consumption ratio of computation and memory access. In their estimation, the energy consumption of SRAM access with the same amount of data is $50 \times$ that of 32-bit addition operations, and the energy consumption of DRAM access is $6400 \times$ that of computations. A theoretical analysis of the required memory shows that the total memory by BoolNet is much lower than previous BNNs, especially during the earlier stages of the network (details are presented in Figure 7b in the appendix). Thus, we expect that our design

| Methods | Bitwidth (W/A/F) | Energy Consumption | Top-1 Acc. | OPs ($\cdot 10^8$) |
|---|---|---|---|---|
| ReActNet (Bi-Real)[†] | 1/1/32 | 3.93mJ | 65.9% | 1.63 |
| Bi-RealNet | 1/1/32 | 3.90mJ | 56.4% | 1.63 |
| XNOR-Net | 1/1/32 | 1.92mJ | 51.2% | 1.59 |
| BoolNet, k=4 (ours) | 1/1/4 | 1.33mJ | 63.0% | 1.64 |
| BaseNet, k=4 (ours) | 1/1/4 | 0.83mJ | 58.2% | 1.54 |
| BaseNet, k=1 (ours) | 1/1/1 | **0.70mJ** | 53.3% | 1.51 |

(a) The advantage of BoolNet is reduced energy consumption.
[†]The ReActNet result based on a Bi-RealNet backbone is stated on the official Github repository (Liu et al., 2020a).

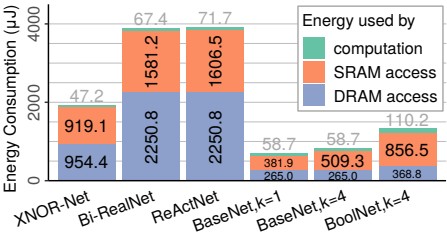

(b) Energy consumption regarding computations and access to DRAM/SRAM.

Figure 4: Comparison between BoolNet and state-of-the-art BNNs. The energy consumption is calculated through hardware simulations.

should achieve much higher energy efficiency due to a lower memory access. Other BNNs store and read 32-bit feature maps in and from memory. On the contrary, BoolNet adopts 1-bit data streams everywhere, except for the accumulation result of convolutions, which is 16-bit. However, because of our Shifted Sign network design, the result can be quantized to 1 bit immediately without cache or DRAM access. This is not as efficiently possible in other BNN designs that use 32-bit aggregation requiring the intermediate 32-bit data to be cached. Due to the small size of on-chip memory, it must be written to DRAM and read out at the next layer. The differences in the bit-width of the data path and the amount of data reading and writing result in a significant gap in energy consumption (highlighted in Figure 8). Our evaluation results (see Figure 4b) show that the energy consumption proportion between computing units and memory access is even more extreme than shown by Yang et al. (2017); Han et al. (2015b). Our interpretation is that the simplified operations of BNNs with the bitwise operations XNOR and popcount further reduces their share of overall energy consumption. This insight shows that memory optimization is more desirable in BNNs compared to 32-bit DNNs to achieve highly energy-efficient AI accelerators.

## 4.5 COMPARISON TO STATE-OF-THE-ART BNNs

We compared our proposed networks to the state-of-the-art BNNs, e.g., ReActNet, Bi-RealNet, XNOR-Net (Liu et al., 2018; 2020b; Rastegari et al., 2016). (For a fair comparison we used the result of a ReActNet based on a Bi-RealNet backbone read from the official Github repository provided by Liu et al. (2020a) which is the same backbone that our network uses.) We found that removing 32-bit elements from previous BNNs, leads to an energy reduction by up to $5.6\times$ (BaseNet with $k$=1), but accuracy drops by 12.6% (see Table 4a). Using the proposed Multi-slice strategy ($k$=4) reduces the accuracy drop by 4.9% and still achieves $4.7\times$ energy reduction. Our BoolNet design further increases the accuracy by 4.8% which overall results in a $2.95\times$ reduction of energy and a 2.9% accuracy drop. Compared to the result of Bi-RealNet (Liu et al., 2018), which has been the basis for other works (Martinez et al., 2020), BoolNet with $k$=4 provides an accuracy improvement of 6.6% (and a $2.9\times$ energy reduction). Overall our results show that our proposed BaseNet and BoolNet can achieve significant energy reduction with little accuracy loss compared to recent state-of-the-art models.

## 5 CONCLUSION

In this paper, we studied how to balance energy consumption and accuracy of binary neural networks. We proposed several simple yet useful strategies to remove or replace 32-bit components from BNNs. Our novel BoolNet with fully binary information flow is constructed and still maintains reasonable accuracy. Experiments on ImageNet and the hardware simulations show that (1) theoretical number of operations does not fully reveal the actual efficiency and (2) BoolNet is more energy-efficient with less computing requirements, lower memory usage and lower energy consumption. We believe this is orthogonal to the goals of previous works and a meaningful first step towards achieving extremely energy-efficient BNNs.

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

## A  APPENDIX

Before we present further details in the following sections, we present an overview on the total amount of computation that was used during this work. We measured the total GPU hours for the three experiments in Section 4.5 of our paper. These experiments (BaseNet k=1, BaseNet k=4, BoolNet k=4) required 192, 256, and 336 GPU hours respectively, in total: 784 GPU hours.

We have recorded more than 7191 GPU hours for our ablation studies and our intermediate, initial, or discarded experiments (some of which were not presented in the paper), but estimate that a further 1500-2000 hours were needed in experiments before we started measuring the GPU runtime.

### A.1 Training Details and Further Experimental Results

The training strategy is mostly based on Bethge et al. (2020). More specifically, we use the RAdam optimizer by Liu et al. (2019) with a learning rate of 0.002 without weight decay, use the *cosine learning rate decay* by Guo et al. (2019), and train with a batch size of 256 for 60 epochs. We only use random flipping and cropping of images to a resolution of $224 \times 224$ for augmentation and finally normalize the data according to the mean and standard deviation of the dataset. During validation we resize the images to $256 \times 256$, and then crop the center with a size of $224 \times 224$ (and normalize in the same manner as during training). Our implementation is based on PyTorch (Paszke et al., 2019), and the code can be found in the supplementary material ZIP archive. The implementations of many previous works can not be sped up with `XNOR` and `popcount` (also observed by Fromm et al. (2020)), since they use padding with zeros, which introduces a third value ($\{-1, 0, +1\}$) in the feature map. To circumvent this issue, we use *Replication* padding, which duplicates the outer-most values of the feature map, thus the values are limited to $\{-1, +1\}$. A further difference to previous work, is our progressive weight binarization technique to remove the need for two-stage trainings, as discussed in the following Section.

#### A.1.1 Progressive Weight Binarization vs. Two-Stage Training

We have introduced the progressive weight binarization strategy in Section 3.4, Equation 7 and discussed the results briefly in Section 4.1. As presented in our main paper, training with progressive weight binarization leads to a higher accuracy, if we train for the same total number of epochs. However, we also conducted an experiment using a linear increase ($\lambda'_t = 1 - t + \epsilon, \epsilon = 10^{-6}$) instead of our proposed exponential increase ($\lambda_t = \sigma^t$) of the slope (see Figure 5). We chose $\sigma$, so the final $\lambda$ values are equal, i.e. if $t_{\max}$ represents the final epoch, then $\lambda_{t_{\max}} = \lambda'_{t_{\max}} = 10^{-6}$. The learning curves show that our progressive weight binarization gains the largest advantage by only "initializing" the values during a brief initial phase of the training.

#### A.1.2 Code Submission

Within the supplementary material ZIP archive we provide our training code. We added all details needed to reproduce each of our experiments depicted in Section 4.5 of our paper in the respective folders:

- BaseNet(k=1) in `BaseNet_k=1`
- BaseNet(k=4) in `BaseNet_k=4`
- BoolNet(k=4) in `BoolNet_k=4`

The complete code used for each run can be found in the subfolder `src` and the exact running command is saved in `src/run.sh` (in some cases the number of GPUs needs to be provided in the environment variable `NUM_GPU` or replaced in the run command).

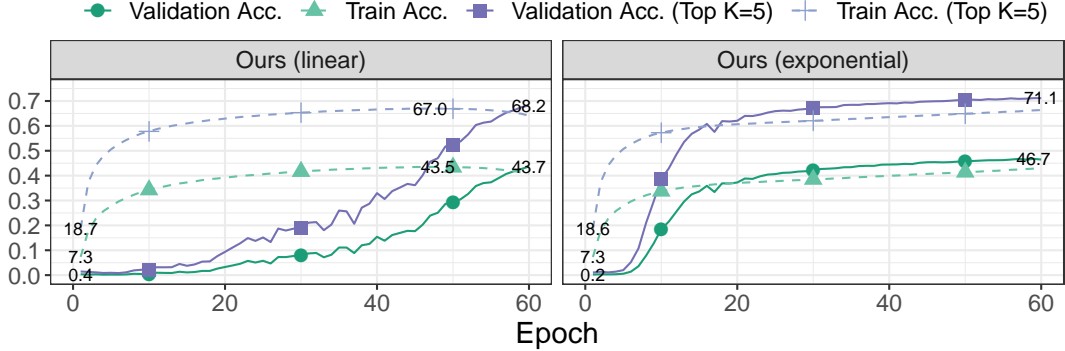

Figure 5: The training and validation accuracy curves of our proposed *Progressive Weight Binarization*. An exponential increase of the slope leads to much better results, than a linear increase.

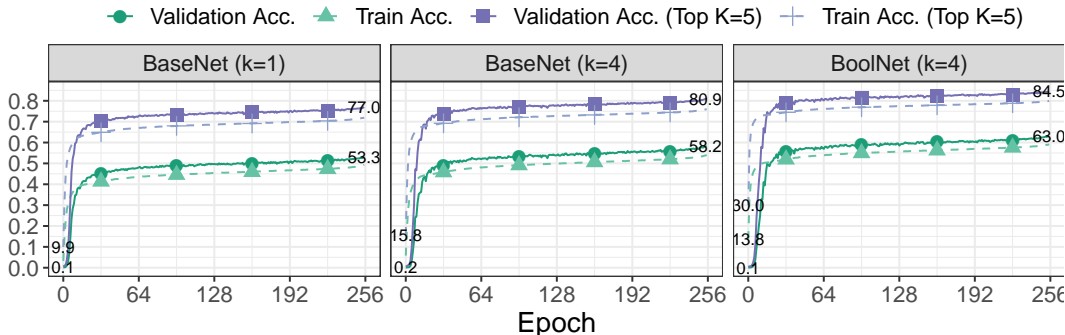

Figure 6: The training and validation accuracy curves of our trainings discussed in the comparison to the state-of-the-art BNNs in Section 4.5.

| | | | AvgPool Acc. (%) | | MaxPool Acc. (%) | | Stride=2 Acc. (%) | |
|---|---|---|---|---|---|---|---|---|
| $k$ | Bits | Groups | Top-1 | Top-5 | Top-1 | Top-5 | Top-1 | Top-5 |
| 1 | 32 | 1 | 63.5 | 87.8 | 63.0 | 87.7 | 60.7 | 86.4 |
| | **1** | 1 | 63.1 | 88.0 | 62.5 | 87.2 | 60.9 | 86.7 |
| 8 | 32 | 1 | 66.0 | 89.4 | 67.0 | 90.0 | 63.4 | 87.9 |
| | 32 | 8 | 65.0 | 88.0 | 65.3 | 88.9 | 62.2 | 87.0 |
| | **1** | 1 | 64.1 | 88.5 | 65.0 | 89.0 | 62.6 | 87.3 |

Table 2: Our ablation study on CIFAR100 regarding different downsampling methods. The number of bits refers to both the input activation and weight binarization of the $1 \times 1$ convolution in the shortcut branch.

We also added the output logs (`logs/training.log`), data CSV (`logs/data.csv`), and total training time (`logs/time`). Further, we plotted the learning curves regarding training and validation accuracy (`logs/acc.png` - a summary of all runs can be seen in Figure 6).

We use a virtualized environment for PyTorch (Paszke et al., 2019) based on Ubuntu for our code setup. The hosts system thus needs support for enroot[1] (or docker[2]). Further, the host system requires the *nvidia-container*[3] library and a recent NVIDIA CUDA driver (we tested driver version 465.27 with CUDA 11.3) for training with GPU. Since several additional requirements are needed to be installed within the Pytorch base image, we provide an example installation script `install.sh`, that can be used to set up the additional requirements inside a virtual container. The `install.sh` script also contains complete commands (in the comments) to setup a container based on enroot. Note, that the ImageNet dataset also needs to be downloaded and prepared manually in the usual manner (using a *train* and *val* folder for the respective split). The validation images need to be moved into labeled subfolders. The dataset then needs be mounted into the container, example commands are in the file `install.sh` (default path inside the container is `/mnt/imagenet/{train,val}`).

## A.2 ABLATION STUDY ON THE DOWNSAMPLE STRUCTURE

As described in Section 3.2, we modify the $1 \times 1$ convolution in the downsampling branch in contrast to many previous works (Rastegari et al., 2016; Liu et al., 2018; 2020b; Martinez et al., 2020). While being helpful for accuracy, the 32-bit $1 \times 1$ convolution involves extra computing, memory and energy consumption, which is in conflict with our motivation. Using our multi-slice strategy with $k = 8$, the number of input channels for the $1 \times 1$ convolution also increases by the same factor of 8. To counter this increase of 32-bit operations, it could be an option to use 8 groups in the convolution, which would keep the number of 32-bit operations constant, compared to previous work. However, this strategy still conflicts with our motivation to remove most 32-bit operations. Furthermore, the average

---

[1] `https://github.com/NVIDIA/enroot`
[2] `https://www.docker.com/`
[3] `https://github.com/nvidia/libnvidia-container`

| Operation | Power (mw) | Area (um²) | Operation | Power (mw) | Area (um²) |
|---|---|---|---|---|---|
| BConv | 108.8 | 131737 | Int8 Conv(1/8) | 504 | 836269 |
| - | - | - | Int Agg | 43.5 | 53238 |
| 16-bit Sign | 1.4 | 7956 | 32-bit Sign | 3.3 | 13548 |
| 32-bit RPReLU | 137.6 | 310671 | Int8 BN | 50.1 | 274606 |

(a) Energy consumption per unit operation and circuit area of commonly used components.

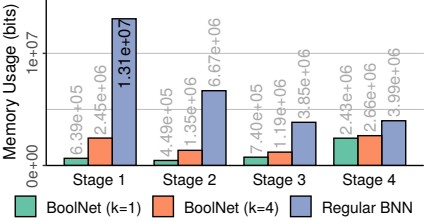

(b) Memory usage comparison between blocks of different stages.

Figure 7: A theoretical memory usage comparison of one convolution block between BoolNet and previous work. Actual numbers can differ during implementation, but BoolNet shows significantly lower memory usage, especially in early stages, even when using our Multi-slice strategy with $k = 4$.

pooling layer used in previous work, requires additional 32-bit addition and division operations, which could be reduced with either using a max pooling layer or a stride of 2.

Therefore, to find a good downsample module with binary data flow, we first design the downsample template as [Conv$_y$, $x$, BN, Sign]. In this template, $x$ indicates the different candidate downsample operations (e.g., average pooling, max pooling, or adding stride=2 to the convolution) and $y$ the number of bits used for weights and activations in the convolution.

We conducted a detailed ablation study on the CIFAR100 dataset for both $k = 1$ and $k = 8$ (see Table 2). The results show, that max pooling combined with 1-bit $1 \times 1$ convolution (groups = 1) has the same Top-1 accuracy as average pooling combined with 32-bit $1 \times 1$ convolution (groups = 8). Thus, we decide to use max pooling instead of average pooling, since it does not involve any 32-bit operations, such as addition and division.

Based on the above analysis, we suggest using the [32-bit Conv (groups = k), AvgPool2d, BN, Sign] structure for the downsample branch if we want to increase accuracy. However, if we intend to build a fully binary data flow, we suggest using the [1-bit Conv (groups = 1), MaxPool2d, BN, Sign] structure (independent of $k$) instead to balance the accuracy and hardware efficiency. The latter is also the structure we used for our experiments in the main paper.

### A.3    More Details About the Energy Consumption Simulation

In Table 3, we give an example of calculating the memory consumption among different stages of our network. Compared with regular BNNs with mixed precision data flow, the fully binary representation of BoolNet significantly lowers the memory consumption during inference process. This change leads to less memory access operations to DRAM which has a much higher power consumption than the on-chip SRAM. To the best of our knowledge, our work is the first one to study the impact of memory access on energy consumption. The details of simulation and energy estimation are introduced as follow.

**Overall hardware architecture**. An illustrative graph on the data flow between the hardware components is provided in Figure 8. In the typical BNN Bi-RealNet, only the convolution is binary, the shortcut branch adopts high precision, and other calculations adopt high precision, too. The corresponding accelerators we designed have different computing modules (but their parallelisms are the same, that is, the computing time of the whole block is roughly the same, and the binary convolution units are exactly the same). In addition, for fair comparison, these accelerators have the same size of on-chip memory (192KB for feature map and 288KB for weight) and the same off-chip memory.

**Computing unit**. The binary convolution units of different BNN accelerators are exactly the same, but other calculation units of BoolNet are simpler. The first is the shortcut branch of downsample blocks. The shortcut branch of traditional BNNs are high-precision, and the high-precision convolution downsampling is adopted. Although the convolution on the shortcut branch accounts only for a small amount of calculation, the power consumption of a high-precision convolution is 37 times that of a binary convolution, and the extra convolution unit also increases the complexity of the circuit.

Table 3: Theoretical minimum memory requirement of all convolution blocks (can differ depending on the implementation). $k$ is the number of slices. The stages have different input size and thus lead to different memory requirements.

| Memory Usage of | Stage 1 with $64 \times 56 \times 56$ | | | Stage 2 with $128 \times 28 \times 28$ | | |
|---|---|---|---|---|---|---|
| | BoolNet (k=1) | BoolNet (k=4) | Regular BNN | BoolNet (k=1) | BoolNet (k=4) | Regular BNN |
| Weights | 36,864 | 36,864 | 36,864 | 147,456 | 147,456 | 147,456 |
| Activation | $200{,}704{\cdot}1$ $= 200{,}704$ | $200{,}704{\cdot}4$ $= 802{,}816$ | $200{,}704{\cdot}1$ $= 200{,}704$ | $100{,}352{\cdot}1$ $= 100{,}352$ | $100{,}352{\cdot}4$ $= 401{,}408$ | $100{,}352{\cdot}1$ $= 100{,}352$ |
| Output & Features | $2{\cdot}200{,}704{\cdot}1$ $= 401{,}408$ | $2{\cdot}200{,}704{\cdot}4$ $= 1{,}605{,}632$ | $2{\cdot}200{,}704{\cdot}32$ $= 12{,}845{,}056$ | $2{\cdot}100{,}352{\cdot}1$ $= 200{,}704$ | $2{\cdot}100{,}352{\cdot}4$ $= 802{,}816$ | $2{\cdot}100{,}352{\cdot}32$ $= 6{,}422{,}528$ |
| **Total** | **638,976** | **2,445,312** | **13,082,624** | **448,512** | **1,351,680** | **6,670,336** |
| Memory Usage of | Stage 3 with $256 \times 14 \times 14$ | | | Stage 4 with $512 \times 7 \times 7$ | | |
| | BoolNet (k=1) | BoolNet (k=4) | Regular BNN | BoolNet (k=1) | BoolNet (k=4) | Regular BNN |
| Weights | 589,824 | 589,824 | 589,824 | 2,359,296 | 2,359,296 | 2,359,296 |
| Activation | $50{,}176{\cdot}1$ $= 50{,}176$ | $50{,}176{\cdot}4$ $= 200{,}704$ | $50{,}176{\cdot}1$ $= 50{,}176$ | $25{,}088{\cdot}1$ $= 25{,}088$ | $25{,}088{\cdot}4$ $= 100{,}352$ | $25{,}088{\cdot}1$ $= 25{,}088$ |
| Output & Features | $2{\cdot}50{,}176{\cdot}1$ $= 100{,}352$ | $2{\cdot}50{,}176{\cdot}4$ $= 401{,}408$ | $2{\cdot}50{,}176{\cdot}32$ $= 3{,}211{,}264$ | $2{\cdot}25{,}088{\cdot}1$ $= 50{,}176$ | $2{\cdot}25{,}088{\cdot}4$ $= 200{,}704$ | $2{\cdot}25{,}088{\cdot}32$ $= 1{,}605{,}632$ |
| **Total** | **740,352** | **1,191,936** | **3,851,264** | **2,434,560** | **2,660,352** | **3,990,016** |

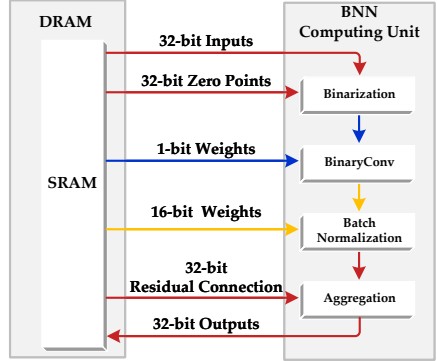
(a) Bi-RealNet Data Flow on Hardware

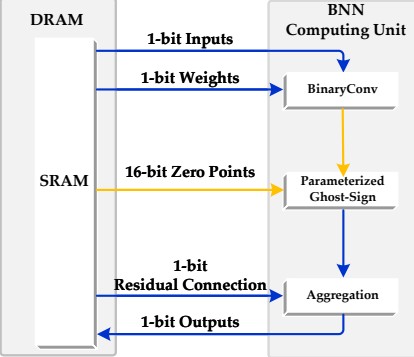
(b) BoolNet Data Flow on Hardware

Figure 8: Hardware data flow comparison between Bi-RealNet and BoolNet.

Secondly, regarding batch normalization and binarization, since the shortcut branch has changed from high-precision to binary, the aggregation position of the shortcut branch and the main branch has also changed, so that the binarization and batch normalization can be simplified together, while the calculation of typical BNN can not be simplified, and their power consumption is high. In addition, there is a difference in the complexity of the aggregation operation itself (boolean logic operation vs. 32-bit addition) and the computational overhead of non-linear functions (i.e. RPReLU) added in networks such as ReActNet. These aspects show the efficiency of BoolNet.

**Energy consumption per unit**. The energy consumption per unit operation of some commonly used components is shown in Figure 7a. However, since the units are not operated the same number of times, the total energy consumption during one inference graph is different. For instance, the energy consumption of Int8 downsampling convolution is $37\times$ larger than binary downsampling[4]. Surprisingly, per unit operation, a 32-bit RPReLU consumes 26% more energy than a binary convolution, Int8 BN consumes about half of a binary convolution, and those two components are commonly used in conjunction with binary convolutions in existing BNNs.

**On-chip memory**. We use CACTI 6.5 to simulate the power of on-chip SRAM. According to the requirements of the computing unit, we configure the on-chip SRAM to meet the parallelism of the corresponding data reading bandwidth (64 bits for BoolNet and 2048 bits for traditional BNNs), while keeping the total storage unchanged. In addition, we split a large SRAM into multiple SRAMs

---

[4]37=504×8/108.8, where Int8 Conv has only 1/8 of the parallel capability of BConv.

to meet the requirement that the read time is less than the clock cycle (1ns) of the computing unit. Finally, the simulation software can give the energy consumption of one read or one write of each SRAM unit. For each layer of the network, we know the total number of operations for each type of operation. According to the circuit parallelism, we can calculate the number of cycles. Then, according to the amount of data that needs to be read from (or written to) SRAM in each cycle, we can get the energy that the accelerator spends to access on-chip SRAM.

**Off-chip memory**. Due to the limited amount of on-chip memory, it is inevitable to save some data to (or read from) off-chip DRAM in BNN computing. In our BoolNet design, due to the large total number of weights, all BNN accelerators need to read weights from DRAM and write to SRAM before the computation of each layer. In addition, for traditional BNN, the intermediate feature maps are larger, which cannot be completely cached on-chip. It is also necessary to save the extra part to DRAM, to read it back in the next layer. With the amount of read-write operations of data to (and from) DRAM and SRAM, the power consumption data of DRAM read-write operations (SRAM has been given by the CACTI simulation in the previous step) is also needed to estimate the overall energy consumption. We use the DDR4 Power Calculator provided by Micron, to configure a DDR UDIMM module composed of four 8Gb x16 chips, which adopts the speed grade of -075E, and the maximum transmission rate is 2666MT/s. The calculator gives the average energy consumption of reading and writing data with 64 bits parallelism.

**Detailed throughput**. The detailed throughput of the accelerators for BaseNet, BoolNet, BiRealNet, ReActNet, XnorNet are 2125, 2044, 2237, 2237, 2237 samples per second, respectively.

