# OpenReview forum: "BoolNet: Streamlining Binary Neural Networks Using Binary Feature Maps"
_ICLR.cc/2022/Conference — ICLR 2022 Submitted_

### Official Review · Reviewer_pRtS · 2021-11-02

**Correctness:** 3
**Technical Novelty And Significance:** 2
**Empirical Novelty And Significance:** 2
**Recommendation:** 3
**Confidence:** 4

**Main Review:**

Strength:

1- Reducing number of FP32 operations in BNN that reduces the energy consumption.

2- The author evaluated the effect of various approaches such as Multi-slice Binary Convolution for BNN on accuracy, memory footprint and number of operations ( the ablation study on ImageNet and ResNet-34 is interesting)

Weakness:

1- The novelty of paper is not obvious. It seems the author uses pervious approaches and combines it with BNN to improve accuracy.

2- The motivation behind using sign function for BN is not explained in the paper. Please elaborate this.

3- In the table on the Figure 4, to have a fair comparison, is distillation and long training epoch considered in pervious works?

4- The author did not compare BooLNet with reference [1] approach which also uses Multi-slice Binary Convolution approach.

[1] Pouransari, Hadi, Zhucheng Tu, and Oncel Tuzel. "Least squares binary quantization of neural networks." Proceedings of the IEEE/CVF Conference on Computer Vision and Pattern Recognition Workshops. 2020.

5- The BooLNet is only evaluated on ImageNet and ResNet-34 and it is not obvious that this proposed work can be generalized for others datasets and networks. I recommend evaluate the BoolNet on the others networks such as MobileNet-V2.

**Summary Of The Paper:**

The paper proposes two novel BNNs (BaseNet and BooLNet) where most of the parameters are represented in the binary formats which is the major difference compared to pervious works where activations in layers are represented in 32-bit floats. Moreover, the proposed networks removes the BN and most of ReLu activations and replaces with sign function. To increase the knowledge capacity, the authors uses Multi-slice Binary Convolution and Local Adaptive Shifting approaches. The inference performance of BaseNet and BooLNet are evaluated and compared with the inference performance of ResNet-34 using 32-bit floating-point format.

**Summary Of The Review:**

As this reviewer mentioned, the motivation to use only binary network is good research direction, However, the novelty of this paper is not obvious and the BoolNet needs to be evaluated on other benchmarks. I recommend this paper to be rejected.

---

> ### Author Response · Authors · 2021-11-14
> **Response to Reviewer pRtS**
>
> >Q1:
> The reviewer mentions in their paper summary: “The paper proposes two novel BNNs where most of the parameters are represented in the binary formats which is the major difference compared to pervious works where activations in layers are represented in 32-bit floats.”
>
> **Answer:** We would like to clarify a misunderstanding in this statement: the parameters are kept in 1-bit in previous works and our work and even the activations (after the sign function) have been 1-bit in previous work and our work as well. Our work attempts to reduce memory consumption by also binarizing the feature maps (which were kept in 32 bits in previous work).
>
> >Q2:
> The reviewer also mentions: “The inference performance of BaseNet and BooLNet are evaluated and compared with the inference performance of ResNet-34 using 32-bit floating-point format.”
>
> **Answer:** There seems to be another misunderstanding here. We did not compare our work to a ResNet-34. Instead it was compared to, e.g., BiRealNet (with 18 layers, which is based on a ResNet-18). Furthermore, as mentioned above, the 32-bit floating point format is only used for the accumulation of the feature maps in previous work. One novel technique of BoolNet is attempting to also keep these feature maps in 1-bit only.
>
> >Q3:
> 1- The novelty of paper is not obvious. It seems the author uses pervious approaches and combines it with BNN to improve accuracy.
>
> **Answer:** We would be grateful if the reviewer could state some specific techniques used in our work, that are also present in previous approaches so we can either clarify the differences or improve the manuscript.
>
> >Q4:
> 2- The motivation behind using sign function for BN is not explained in the paper. Please elaborate this.
>
> **Answer:** The motivation behind combining the sign function with the BN layer is to reduce the number of required operations. This allows 32-bit floating point multiplication/division operations to be removed from the network.
>
> >Q5:
> 3- In the table on the Figure 4, to have a fair comparison, is distillation and long training epoch considered in pervious works?
>
> **Answer:** As mentioned in our text, we actually use the distillation approach that is also used in ReActNet. Their training time is even longer (512 epochs in total) than ours (256 epochs). Furthermore, their code [1] shows additional data augmentation techniques (Lighting augmentation) not mentioned in their work, which were not used in BoolNet.
>
> BiRealNet was trained for 256 epochs, but did not use distillation. In our experience knowledge distillation can add about 2% of accuracy, which would still be lower than BoolNet with a higher energy consumption.
>
> To the best of our knowledge XNOR-Net was trained for about 60 epochs and without knowledge distillation.
>
> >Q6:
> 4- The author did not compare BooLNet with reference [1] approach which also uses Multi-slice Binary Convolution approach.
> [1] Pouransari, Hadi, Zhucheng Tu, and Oncel Tuzel. "Least squares binary quantization of neural networks." Proceedings of the IEEE/CVF Conference on Computer Vision and Pattern Recognition Workshops. 2020.
>
> **Answer:** We thank the reviewer for their suggestion of related work. However, it seems the suggested work does not use multiple slices. They propose a quantization for k bits instead, which is different from applying different zero points to achieve the multiple slices as proposed in our work.
>
> The result of their quantization approach used with k-bits does seem dependent on the input values and each input value is combined with each other value in the convolution, which leads to a higher number of k-bit operations. In our method, by using a grouped convolution, we avoid using higher-bit operations and reduce the amount of computation instead of increasing it.
>
> >Q7:
> 5- The BooLNet is only evaluated on ImageNet and ResNet-34 and it is not obvious that this proposed work can be generalized for others datasets and networks. I recommend evaluate the BoolNet on the others networks such as MobileNet-V2.
>
> **Answer:** We did not use ResNet-34. The models we used are similar to a ResNet 18 but with a double skip connection (often called BiRealNet instead).
>
> Other previous work also only used ImageNet and a single architecture (often ResNet-style) to evaluate their approach. How can we compare to these previous works, if we choose a different base architecture?
>
> [1] Github Implementation of ReActNet, https://github.com/liuzechun/ReActNet/blob/master/resnet/1_step1/train.py#L118

---

> > ### Comment · Reviewer_pRtS · 2021-11-30
> > **Response to Authors**
> >
> > Thank you for the additional comments and clarifications. I appreciate the time the authors and the reviewers spent to discuss this manuscript. I will keep my current score.

---

### Official Review · Reviewer_nRTL · 2021-11-02

**Correctness:** 3
**Technical Novelty And Significance:** 4
**Empirical Novelty And Significance:** 3
**Recommendation:** 8
**Confidence:** 5

**Main Review:**

Strengths :
+ The first work for fully 1bit neural networks, which is very important for practical deployment on hardware.
+ The effects of 32-bit layers in commen BNNs are analyzed and removed by specific design.
+ A Multi-slice strategy and other tricks are proposed to alleviate the accuracy loss.
+ State-of-the-art performance on the trade-off between accuracy and energy consumption.

Weaknesses:
- The introduced MS-BConv increases the number of channels. Will it increase the memory usage and inference time?
- MS-BConv increases the number of channels, which is related to [1]. This related work should be included.
[1] Searching for accurate binary neural architectures. ICCVW 2019.

**Summary Of The Paper:**

The paper is the first to build fully 1bit neural networks. The proposed BoolNet achieves state-of-the-art performance on the trade-off between accuracy and energy consumption.

**Summary Of The Review:**

This paper is the first work to build fully 1bit neural networks. I like this work and recommend acceptance for it.

---

> ### Author Response · Authors · 2021-11-14
> **Response to Reviewer nRTL**
>
> We thank the reviewer for their valuable time and their feedback. We are happy to see their approval but also would like to elaborate on the two issues the reviewer has raised:
>
> >Q1:
> The introduced MS-BConv increases the number of channels. Will it increase the memory usage and inference time?
>
> **Answer:** Using MS-BConv (k > 1) leads to an increase of (binary) operations (see lower half of Table 1) and memory consumption (see Table 3 in the appendix) compared to using k = 1.
> However, BoolNet with k=4 still uses lower memory consumption (4 bits per value in the feature map) than regular BNNs (32 bits per value). This is also shown in Table 3.
> The inference speed measured in our simulation is similar between all networks, since different inference speeds would complicate a fair energy consumption comparison.
>
> >Q2:
> MS-BConv increases the number of channels, which is related to [1]. This related work should be included. [1] Searching for accurate binary neural architectures. ICCVW 2019.
>
> **Answer:** We thank the reviewer for the suggestion and agree it would be suitable to add this work to our related work section.

---

### Official Review · Reviewer_MNqE · 2021-11-02

**Correctness:** 3
**Technical Novelty And Significance:** 2
**Empirical Novelty And Significance:** 2
**Recommendation:** 3
**Confidence:** 2

**Main Review:**

This reviewer has several concerns about this paper as below:
- This reviewer is curious about the portion of overhead due to full-precision layers during BNN inference. This paper refers to a previous study (Fromm et al., 2020), but in that paper, only squeezenet was evaluated. To strengthen the motivation of this paper, this paper also includes simulation results on energy consumption, but it is a little hard for this reviewer to accept the simulated results because 1) a few networks are compared with the proposed work and 2) simulation results strongly depend on assumptions about h/w implementations. BaseNet/BoolNet with 32-but features can be comparable networks for ablation study to show the overhead of 32-bit features. Since current BNNs show better accuracy than the proposed work, this paper should strengthen and prove the motivation, why we need to eliminate the 32-bit features from BNNs completely, with deeper analysis.
- This paper claims that existing BNNs use various 32-bit features and proposes various techniques to make neural networks that consist of only binary operations. However, some ideas seem to be similar to previous works. In addition, for some ideas, there seems to be a lack of comparisons and citations.
  - The proposed shifted sign idea looks similar to ‘Batchnorm-activation as Threshold’ (Umuroglu et al., 2017). which is not cited and compared. This idea is not surprised for BNN researchers.
  - Multi-slice binary convolution layers seem to make a significant impact on improved accuracy. But this idea looks similar to ABC-net (Lin, 2017) and GroupNet (Zhuang, 2019). While ABC-net is compared with some descriptions, the GroupNet (it may be improved than ABC-net) is not compared and it is simply mentioned just one time. This reviewer wants to know why GroupNet is not compared.
  - Figure 4 uses ReActNet(Bi-Real-based) for comparison. Why don’t you use ReActNet-A that shows better accuracy and lower FLOPS? Indeed, the GitHub repo of ReActNet also includes ReActNet-A.

- Minor
  - Page 4 : with an running → with a running

Umuroglu, Yaman, et al. "Finn: A framework for fast, scalable binarized neural network inference." Proceedings of the 2017 ACM/SIGDA International Symposium on Field-Programmable Gate Arrays. 2017.

Lin, Xiaofan, Cong Zhao, and Wei Pan. "Towards accurate binary convolutional neural network." arXiv preprint arXiv:1711.11294 (2017).

Zhuang, Bohan, et al. "Structured binary neural networks for accurate image classification and semantic segmentation." Proceedings of the IEEE/CVF Conference on Computer Vision and Pattern Recognition. 2019.

**Summary Of The Paper:**

This paper aims to eliminate 32bit features of BNNs as possible. This paper claims that existing BNNs embed 32bit features, which can improve the accuracy of BNNs but must lead to overheads during inference. In the proposed network, full-precision batchnorm layers are replaced with 'shifted-sign' layers, full-precision scaling factors are eliminated, and multi-slice binary convolutions and 1b shortcuts are used. The network achieves 63% accuracy on ImageNet with various techniques and knowledge distillation.

**Summary Of The Review:**

As this reviewer mentioned, a more concrete analysis for the motivation should be added and comparisons with previous BNN architectures are required. The current manuscript seems to be lack novelty in this reviewer’s opinion. But, this reviewer is ready to listen to other reviewers’ opinions.

---

> ### Author Response · Authors · 2021-11-14
> **Response to Reviewer MNqE (3/3)**
>
> >Q5:
> Multi-slice binary convolution layers seem to make a significant impact on improved accuracy. But this idea looks similar to ABC-net (Lin, 2017) and GroupNet (Zhuang, 2019). While ABC-net is compared with some descriptions, the GroupNet (it may be improved than ABC-net) is not compared and it is simply mentioned just one time. This reviewer wants to know why GroupNet is not compared.
>
> **Answer:** We thank the reviewer for the suggestion and agree, a mention of GroupNet would be helpful to better understand the context. We adapted the subsection “Multi-slice Binary Convolution” in Section 3.3 accordingly and included other works, such as GroupNet [6] and BNN-Ensemble [7], in the discussion.
>
> However, GroupNet (in a similar way as ABC-Net and BNN-Ensemble) has a different motivation (and implementation) compared to our proposed Multi-slice binary convolution.
> GroupNet increases representational capacity (using 32-bit feature maps as it is based on BiRealNet) with additional weights and inputs (with a certain factor k). Thus, it significantly increases hardware requirements which is directly opposite of the motivation of BoolNet.
> The theoretical numbers such as FLOPs and model size also highlight this, concrete numbers for GroupNet with k=5 on ImageNet are mentioned in [5]: 67% accuracy, but it needs 9.2MB model size and uses 2.68x10^8 operations, significantly more than Bi-RealNet and BoolNet.
>
> However, our proposed Multi-slice binary convolution instead attempts to compress the representation of a 32-bit feature map into a composition of k 1-bit values to reduce the required memory.
>
> >Q6:
> Figure 4 uses ReActNet(Bi-Real-based) for comparison. Why don’t you use ReActNet-A that shows better accuracy and lower FLOPS? Indeed, the GitHub repo of ReActNet also includes ReActNet-A.
>
> **Answer:** We would like to invite the reviewer to read our general statement (posted as a top level comment) about the peculiarities of the OPs reduction in ReActNet-A. It is mostly achieved by their choice of MobileNet as a base model instead of our base model, a ResNet. Applying the ideas presented in our work to a MobileNet architecture would (independently of the actual new methods) directly lead to a major reduction in OPs as well.
>
> Unfortunately, designing and simulating multiple completely different hardware accelerators (because of the inherent underlying difference in architecture) for networks based on MobileNet is not in the scope of this single work and complicates achieving a fair comparison with other networks.
>
> >Q7:
> Minor - Page 4 : with an running → with a running
>
> **Answer:** We thank the reviewer for spotting this error and we corrected it in our source files.
>
> **References:**
>
> [1] Bannink, Tom, Adam Hillier, Lukas Geiger, Tim de Bruin, Leon Overweel, Jelmer Neeven, and Koen Helwegen. "Larq Compute Engine: Design, Benchmark and Deploy State-of-the-Art Binarized Neural Networks." Proceedings of Machine Learning and Systems 3 (2021).
>
> [2] Martinez, Brais, Jing Yang, Adrian Bulat, and Georgios Tzimiropoulos. "Training binary neural networks with real-to-binary convolutions." In International Conference on Learning Representations. 2019.
>
> [3] Bulat, Adrian, Brais Martinez, and Georgios Tzimiropoulos. "High-capacity expert binary networks." In International Conference on Learning Representations. 2021.
>
> [4] Liu, Zechun, Baoyuan Wu, Wenhan Luo, Xin Yang, Wei Liu, and Kwang-Ting Cheng. "Bi-real net: Enhancing the performance of 1-bit cnns with improved representational capability and advanced training algorithm." In Proceedings of the European conference on computer vision (ECCV), pp. 722-737. 2018.
>
> [5] Bethge, Joseph, Christian Bartz, Haojin Yang, Ying Chen, and Christoph Meinel. "MeliusNet: An Improved Network Architecture for Binary Neural Networks." In Proceedings of the IEEE/CVF Winter Conference on Applications of Computer Vision, pp. 1439-1448. 2021.
>
> [6] Zhuang, Bohan, Chunhua Shen, Mingkui Tan, Lingqiao Liu, and Ian Reid. "Structured binary neural networks for accurate image classification and semantic segmentation." In Proceedings of the IEEE/CVF Conference on Computer Vision and Pattern Recognition, pp. 413-422. 2019.
>
> [7] Zhu, Shilin, Xin Dong, and Hao Su. "Binary ensemble neural network: More bits per network or more networks per bit?." In Proceedings of the IEEE/CVF Conference on Computer Vision and Pattern Recognition, pp. 4923-4932. 2019.

---

> ### Author Response · Authors · 2021-11-14
> **Response to Reviewer MNqE (2/3)**
>
> >Q3:
> BaseNet/BoolNet with 32-but features can be comparable networks for ablation study to show the overhead of 32-bit features. Since current BNNs show better accuracy than the proposed work, this paper should strengthen and prove the motivation, why we need to eliminate the 32-bit features from BNNs completely, with deeper analysis.
>
> **Answer:** Our main motivation in this work is to reduce energy consumption and achieve higher efficiency in this area. Is the reviewer not convinced that this motivation is desirable or are they not convinced this work is achieving a meaningful step into this direction? Our results show that a 2.95x reduction in energy consumption is achievable.
>
> >Q4:
> This paper claims that existing BNNs use various 32-bit features and proposes various techniques to make neural networks that consist of only binary operations. However, some ideas seem to be similar to previous works. In addition, for some ideas, there seems to be a lack of comparisons and citations.
> The proposed shifted sign idea looks similar to ‘Batchnorm-activation as Threshold’ (Umuroglu et al., 2017). which is not cited and compared. This idea is not surprised for BNN researchers.
>
> **Answer:** We were not familiar with the proposed work and thank the reviewer for suggesting this related work.
> We checked the details of the implementation, and although the mathematical formulation is indeed similar to our proposed Shifted Sign, the ‘Batchnorm-activation as Threshold’ can not be applied directly to recent works. We cite from (Umuroglu et al., 2017):
> All BNN layers use batch normalization [11] on convolutional or fully connected layer outputs, then apply the sign function to determine the output activation.
>
> This assumption does not hold true for recent BNN works, such as Bi-RealNet, ReActNet, etc., anymore, as these works use a different layer order in the block structure (Sign -> Conv -> BN) and accumulate features in a 32-bit feature map in between blocks with shortcuts. This idea of explicitly using 32-bit feature maps to strengthen the representative capacity of BNNs as presented by Bi-RealNet [4] has been used in many works afterwards and it prohibits using the proposed ‘Batchnorm-activation as Threshold’.
>
> Thus we argue that only by rethinking the need to accumulate features as 32-bit values to using 1-bit features in our work, ideas such as ‘Batchnorm-activation as Threshold’ and Shifted Sign can be used again. Consequently, we added the proposed reference to our draft in Section 3.2 and pointed out the similarity in implementation but also the necessity of restructuring the network in our proposed way.

---

> ### Author Response · Authors · 2021-11-14
> **Response to Reviewer MNqE (1/3)**
>
> We thank the reviewer for their valuable time and their feedback. We would like to comment on some of the issues the reviewer has raised and hope we can continue the discussion if the reviewer still has concerns about these issues:
>
> >Q1:
> This reviewer is curious about the portion of overhead due to full-precision layers during BNN inference. This paper refers to a previous study (Fromm et al., 2020), but in that paper, only squeezenet was evaluated.
>
> **Answer:** We encourage the reviewer to look at another very recent work by Bannink et al. [1] that has investigated this specific overhead and proposes ideas to reduce the overhead of 32-bit operations. They investigate many different network types, e.g., BiRealNet, Real-To-Binary, BinaryDenseNet, MeliusNet, XNOR-Net.
>
> Although they do not study energy consumption and still use 32-bit feature maps, their work proves the significance of removing 32-bit operations as much as possible from BNNs for achieving high inference speeds. We also added the work [1] to our introduction and related work section.
>
> >Q2:
> To strengthen the motivation of this paper, this paper also includes simulation results on energy consumption, but it is a little hard for this reviewer to accept the simulated results because 1) a few networks are compared with the proposed work and 2) simulation results strongly depend on assumptions about h/w implementations.
>
> **Answer:** Regarding point 2): We would like to refer to our response eCWL:Q3, which is very related (the reviewer should be able to quickly find the comment by searching for this code). We hope they find this answer satisfactory. Otherwise we hope they can provide some more insight on which part of our process or assumptions they think could skew the results so we can try to clear up any potential misunderstandings.
>
> Regarding point 1): Reimplementing the networks faithfully and correctly is very time-consuming, since the hardware simulation and energy calculation depends on knowing the network details (and previous works do not always publish source code). Thus, we tried to select a range of networks from “traditional” approaches (XNOR-Net) to very recent state-of-the-art ones (ReActNet based on Bi-Real) for comparison.
>
> We are interested in why the reviewer does not think the selected networks pose a suitable comparison. We hope the reviewer could propose networks that are more suitable in their mind and include their reasoning for why those networks should be included in the comparison.
>
> We found that many previous works, for example [2,3,4] are based on Bi-RealNet, but some of them only introduce novel training techniques. Thus, by including BiRealNet the performance of these approaches, which build on BiRealNet but do not add additional network elements, can also be measured.

---

### Official Review · Reviewer_eCWL · 2021-11-02

**Correctness:** 2
**Technical Novelty And Significance:** 3
**Empirical Novelty And Significance:** Not applicable
**Recommendation:** 5
**Confidence:** 4

**Main Review:**

This paper is an attempt to reduce the precision of 32-bit components in BNNs. To this end, a new network architecture is introduced that integrates batchnorm into sign function, modifies downsample block and skip connections to reduce their required precision, and exploit a multi-slice convolution to compensate for the loss incurred by the precision reduction techniques.

Strengths:
- The paper is well-written and easy to understand.
- The proposed network architecture is novel and is an attempt to further reduce the complexity of neural networks for their efficient deployment.
- There is a discussion on one employment scenario and its implementation results to support the effectiveness of the proposed architecture.

Weaknesses:
- My first concern is the accuracy performance of the proposed architecture w.r.t. the SOTA architecture (i.e., ReActNet). In the ReActNet paper, there is a network architecture called ReActNet-A which achieves the accuracy of 69.4% while requiring 0.87 OPs. If we compare performance of this architecture given the accuracy and the number of OPs with the results reported in Figure 4(a), we can see ReActNet-A significantly outperforms BoolNets and BaseNets.
- My second concern is the implementation results of BoolNets. In industry, we don't usually design a custom hardware for a specific network architecture. On the other hand, we consider a general custom hardware that can support a wide range of computing units and layer types. Of course, in this scenario, each network architecture will show a different implementation performance on the hardware accelerator. Therefore, I believe the hardware performance of BoolNets and ReActNets should be measured on a specific hardware accelerator for a fair comparison, and the choice of hardware accelerator should be limited to well-know ones that are being used commercially. Otherwise, the design of custom hardware for a specific network can be biased to favor that network and does not constitute a fair comparison in my opinion.


**Summary Of The Paper:**

This paper proposes methods to further reduce energy consumption of binary neural networks by removing or replacing 32-bit components (e.g., skip connections) in SOTA BNNs. More specifically, the proposed architecture (1) reduces the precision of skip connections and activation functions, (2) transforms the BatchNorm layer into a simple sign function, and (3) employs a multi-slice strategy to alleviate the loss of representational capacity incurred by binarizing the feature maps and shortcut connections. The results shows that the new model achieves 4.7x energy reduction with some accuracy degradation compared to SOTA architectures.

**Summary Of The Review:**

After carefully reading the paper, I believe while this paper made novel contributions in reducing the precision of 32-bit component of BNNs, it couldn't show its advantages over existing works such as ReActNet. More specifically, ReActNet-A achieves a better accuracy and also requires a lower number of OPs, making the advantage of BoolNet limited to the implementation results. However, the implementation results on a custom hardware designed for a specific network doesn't establish a fair comparison. My suggestion is to report hardware performance of the networks on a commercial accelerator instead for a fair comparison and more reliable results.

---

> ### Author Response · Authors · 2021-11-14
> **Response to Reviewer eCWL (2/2):**
>
> >Q3:
> Otherwise, the design of custom hardware for a specific network can be biased to favor that network and does not constitute a fair comparison in my opinion.
>
> **Answer**: We agree that such designs can be biased to favor a network, but we strictly used the same conditions and parameters for the simulation of all accelerators. For reference we cite the second paragraph of section 4.4 below our answer [7].
> Which of these decisions does the reviewer suspect could lead to an unfairness? How familiar is the reviewer with the process of hardware simulation and the energy calculation? We have not developed the whole process in this work, it is based on the references provided in our work [1,2].
>
> The general process of showing new potential through hardware simulation is also widely used in the hardware development community. For example, [3,4,5] in the latest HPCA 2021 conference also get the power results of computing units with post synthesis simulation, get the read and write energy of SRAM with CACTI, and get the DRAM access energy with other DRAM simulation software. In our work we used the same way to do our hardware experiments.
>
> If the reviewer is interested in more details or has certain design decisions in mind that could skew the performance between the examined networks, we would be happy to continue the discussion and provide additional details. The reviewer can also refer to the supplementary material, which contains more details than the main work regarding the hardware simulation.
>
> **References:**
>
> [1] Conti, Francesco, Pasquale Davide Schiavone, and Luca Benini. "XNOR neural engine: A hardware accelerator IP for 21.6-fJ/op binary neural network inference." IEEE Transactions on Computer-Aided Design of Integrated Circuits and Systems 37, no. 11 (2018): 2940-2951.
>
> [2] Zhang, Yichi, Junhao Pan, Xinheng Liu, Hongzheng Chen, Deming Chen, and Zhiru Zhang. "FracBNN: Accurate and FPGA-efficient binary neural networks with fractional activations." In The 2021 ACM/SIGDA International Symposium on Field-Programmable Gate Arrays, pp. 171-182. 2021.
>
> [3] Wang, Hanrui et al. “SpAtten: Efficient Sparse Attention Architecture with Cascade Token and Head Pruning.” 2021 IEEE International Symposium on High-Performance Computer Architecture (HPCA) (2021): 97-110.
>
> [4] Jiang, Lei and Farzaneh Zokaee. “EXMA: A Genomics Accelerator for Exact-Matching.” 2021 IEEE International Symposium on High-Performance Computer Architecture (HPCA) (2021): 399-411.
>
> [5] Li, Jiajun et al. “CSCNN: Algorithm-hardware Co-design for CNN Accelerators using Centrosymmetric Filters.” 2021 IEEE International Symposium on High-Performance Computer Architecture (HPCA) (2021): 612-625.
>
> [6] Bethge, Joseph, Christian Bartz, Haojin Yang, Ying Chen, and Christoph Meinel. "MeliusNet: An Improved Network Architecture for Binary Neural Networks." In Proceedings of the IEEE/CVF Winter Conference on Applications of Computer Vision, pp. 1439-1448. 2021.
>
> [7] Cite from our work, section 4.4:
>
> We designed the five accelerators in the RTL language. Then, the power and area of computing circuits is given by Design Compiler (DC) with a TSMC 65nm process and 1GHz clock frequency. We refer to the design and implementation methods of computing units of Conti et al. (2018); Zhang et al. (2021).
> For a fair comparison between the different BNNs, we keep the design of architecture, data stream, the parallelism of computing units, and total on-chip cache (192KB for feature maps and 288KB for weights) consistent and only change the bit-width of the data stream and computing units.
> More specifically, the parallelism of binary convolution is 64x64, and the parallelism of other units is 64 in all accelerators (except the IntConv module is 8x64). These modules are pipelined and run at 1GHz.
> When DRAM bandwidth can be fully utilized, the performance depends on the parallelism and is bounded by the convolution, so each accelerator has the same peak performance for convolution, i.e., 4096 GOPs/s.
> Therefore, we achieve similar throughputs between 2044 and 2237 samples per second and it is reasonable to compare the energy consumption of the whole inference process.

---

> ### Author Response · Authors · 2021-11-14
> **Response to Reviewer eCWL (1/2):**
>
> We thank the reviewer for their valuable time and their feedback. We would like to comment on some of the issues the reviewer has raised and hope we can continue the discussion if the reviewer still has concerns about these issues:
>
> >Q1:
> My first concern is the accuracy performance of the proposed architecture w.r.t. the SOTA architecture (i.e., ReActNet). In the ReActNet paper, there is a network architecture called ReActNet-A which achieves the accuracy of 69.4% while requiring 0.87 OPs. If we compare performance of this architecture given the accuracy and the number of OPs with the results reported in Figure 4(a), we can see ReActNet-A significantly outperforms BoolNets and BaseNets.
>
> **Answer**: We would like to invite the reviewer to read our general statement (posted as a top level comment) about the peculiarities of the OPs reduction in ReActNet-A. It is mostly achieved by their choice of **MobileNet** as a base model instead of our base model, a **ResNet**. Applying the ideas presented in our work to a **MobileNet** architecture would (independently of the actual new methods) directly lead to a major reduction in OPs.
>
> Unfortunately, designing and simulating multiple completely different hardware accelerators (because of the inherent underlying difference in architecture) for networks based on MobileNet is not in the scope of this single work and further reduces comparability.
>
> >Q2
> My second concern is the implementation results of BoolNets. In industry, we don't usually design a custom hardware for a specific network architecture. On the other hand, we consider a general custom hardware that can support a wide range of computing units and layer types. Of course, in this scenario, each network architecture will show a different implementation performance on the hardware accelerator. Therefore, I believe the hardware performance of BoolNets and ReActNets should be measured on a specific hardware accelerator for a fair comparison, and the choice of hardware accelerator should be limited to well-know ones that are being used commercially.
>
> **Answer**: Since our idea is inherently based on a hardware-related improvement (reducing memory consumption), how can we prove the advantage of our approach without also using those advantages of reduced hardware requirements to achieve more efficient hardware?
>
> To us this seems difficult to show in any other way, but we would be grateful if the reviewer can offer some more insight. To present an analogy: to optimize the energy consumption of the RAM in a regular PC, it does not help to look at the fill percentage of RAM, since both powering the RAM chip and read/write operations are the main energy consumers, regardless of the amount of data stored. Although a certain advantage can be seen by reducing the number of read-write operations, we can only observe the full potential of energy saving if we install only half the amount of RAM. Whether that actually means, every user is using half the amount of RAM is then another separate choice (the user might not be interested in the highest efficiency at all). Or in other words: only by using the most efficient/suitable accelerator for a task can we measure the true efficiency potential.
>
> We understand the desire of the reviewer and agree that using a generally available common accelerator for BNNs could also show a certain advantage in a fair way (albeit with reduced potential). However, to the best of our knowledge, there is no commonly used (and commercially available) accelerator for BNNs yet. Which accelerator does the reviewer have in mind that would be suitable? (Our answer A3 also contains more details on the measures we have taken to present a fair comparison.)
>
> Furthermore, we believe our work (and similar ongoing works in the present and future) should and need to lay the groundwork for the first designs of such commercially available accelerators for BNNs, since currently the common ground is not clear (i.e. designing the first “custom but general hardware” the reviewer is talking about). For example, should 32-bit feature maps be supported (such as used in previous work)? Or should such an accelerator support only 1-bit feature maps (such as presented in our work) for the highest energy-efficiency? We think the industry and the community have to decide this after both options have been researched, but we believe this “decision” has not been made, yet.
>
> Thus, we would like to contribute our idea as a new alternative to previous approaches to highlight and maximize the unique potential of BNNs: energy-efficiency (albeit it currently is based on the assumption of using an accelerator design similar to our simulated one).

---

### Author Response · Authors · 2021-11-14
**General Answer (2/2)**

Nevertheless, we conducted additional comparative experiments between BoolNet and ReActNet-A on the MobileNet-backbone. Due to time constraints, we cannot complete the accelerator design and power consumption comparison in a short time. However, we believe that the results based on the ResNet-backbone in the paper have proven BoolNet's energy consumption advantages. Therefore, the additional results focus on the comparison of accuracy with the aligned computational complexity. ​​Based on the results of CIFAR-100, the accuracy gap between BoolNet and ReActNet-A is only 0.72%. However, BoolNet shows 2.95 times more energy-saving than ReActNet in our paper.

|   Method   | CIFAR-100 Top-1 Accuracy (%) | ImageNet Top-1 Accuracy (%) |
|:----------:|:----------------------------:|:---------------------------:|
| ReActNet A |             65.8             |             69.4            |
|   BoolNet  |             65.08            |           running           |

Some details:
For the CIFAR 100 dataset, we based our code on the official ReActNet code on Github. We applied the original two-stage training method used in ReActNet and our proposed progressive training method for BoolNet. Other hyper parameters such as epochs (512) and data augmentations remain the same.
The experiment on ImageNet is currently still running, and should be available in approximately 6 days.

**References**

[1] Bannink, Tom, Adam Hillier, Lukas Geiger, Tim de Bruin, Leon Overweel, Jelmer Neeven, and Koen Helwegen. "Larq Compute Engine: Design, Benchmark and Deploy State-of-the-Art Binarized Neural Networks." Proceedings of Machine Learning and Systems 3 (2021).

[2] Liu, Zechun, Baoyuan Wu, Wenhan Luo, Xin Yang, Wei Liu, and Kwang-Ting Cheng. "Bi-real net: Enhancing the performance of 1-bit cnns with improved representational capability and advanced training algorithm." In Proceedings of the European conference on computer vision (ECCV), pp. 722-737. 2018.

[3] Martinez, Brais, Jing Yang, Adrian Bulat, and Georgios Tzimiropoulos. "Training binary neural networks with real-to-binary convolutions." In International Conference on Learning Representations. 2019.

[4] Rastegari, Mohammad, Vicente Ordonez, Joseph Redmon, and Ali Farhadi. "Xnor-net: Imagenet classification using binary convolutional neural networks." In European conference on computer vision, pp. 525-542. Springer, Cham, 2016.

[5] Liu, Zechun, Zhiqiang Shen, Marios Savvides, and Kwang-Ting Cheng. "Reactnet: Towards precise binary neural network with generalized activation functions." In European Conference on Computer Vision, pp. 143-159. Springer, Cham, 2020.

[6] Bethge, Joseph, Christian Bartz, Haojin Yang, Ying Chen, and Christoph Meinel. "MeliusNet: An Improved Network Architecture for Binary Neural Networks." In Proceedings of the IEEE/CVF Winter Conference on Applications of Computer Vision, pp. 1439-1448. 2021.

---

### Author Response · Authors · 2021-11-14
**General Answer (1/2)**

We highly thank all the reviewers for their time and valuable feedback and would like to address one of the more general issues.

We are happy to find that most of the reviewers understand our main idea in this work: lowering the already low energy consumption of BNNs to extreme levels while maintaining a reasonable level of accuracy. The reviewers also seem to have a high interest in a comparison to ReActNet-A [5], which we completely understand, but think this desire is based on two misunderstandings:

(1) The number of operations alone is not a suitable metric for the efficiency of a network (regarding energy consumption and even runtime) unless a similar topology is used, which is discussed and empirically proven in [1]. Therefore, we chose to build our work upon the well-known and more widely applied (e.g. in [2,3,4]) basis of a **ResNet** with double skip connections (in the original work [2] referred to as Bi-RealNet). However, ReActNet-A is based on **MobileNet**, which inherently has a lower number of FLOPS, which also leads us to the second misunderstanding.

(2) Because ReActNet-A (compared to works based on ResNet or Bi-RealNet [2,3,4] and our proposed BoolNet/BaseNet) uses **MobileNet**, the first convolution is replaced and “artificially” (by this we mean it is unrelated to their proposed method) lowers the number of operations, which could potentially mislead a reader of their work. If we apply the same replacement of the first convolution to a BiRealNet the network FLOPs are reduced from 1.63 (*10^8) to 0.68 (*10^8).

This is because the first convolution alone in a regular **ResNet** based BNN represents **65%** of the number of operations (since it uses a 7x7 kernel and is usually kept in 32-bit in previous work, again [2,3,4] are some examples). This fact is stated in [6] and indirectly also discussed in [1].

Therefore, ReActNet [5] (and other work, such as MeliusNet [6] and QuickNet [1]) gain a large advantage on the theoretical number of operations, simply by replacing the first layer with a different structure. By lowering the number of operations required in the first layer, these works [1,5,6] immediately gain a “FLOPs” advantage before applying any of their methods. While this change is thoroughly discussed in [1] and [6], it could be argued that the authors of [5] did not explain this difference very well, but we do not want to judge the omission of such details in previous work.

We hope the reviewers agree that because of these two issues the comparison to ReActNet-A and only focusing on FLOPs as a single metric for efficiency is not a fair comparison at all, which was also shown in [1].

In our work we do not want to distract from our actual method, by introducing a replacement of the first layer (other replacement options are discussed in [1] and [6]). Designing the hardware accelerator for different base architectures also decreases comparability.

Therefore, we compared BaseNet and BoolNet to well-performing works based on the ResNet architecture and (because of issue number (1)) include actual hardware simulation to measure energy-efficiency instead of simply relying on theoretical counting.

**References**

[1] Bannink, Tom, Adam Hillier, Lukas Geiger, Tim de Bruin, Leon Overweel, Jelmer Neeven, and Koen Helwegen. "Larq Compute Engine: Design, Benchmark and Deploy State-of-the-Art Binarized Neural Networks." Proceedings of Machine Learning and Systems 3 (2021).

[2] Liu, Zechun, Baoyuan Wu, Wenhan Luo, Xin Yang, Wei Liu, and Kwang-Ting Cheng. "Bi-real net: Enhancing the performance of 1-bit cnns with improved representational capability and advanced training algorithm." In Proceedings of the European conference on computer vision (ECCV), pp. 722-737. 2018.

[3] Martinez, Brais, Jing Yang, Adrian Bulat, and Georgios Tzimiropoulos. "Training binary neural networks with real-to-binary convolutions." In International Conference on Learning Representations. 2019.

[4] Rastegari, Mohammad, Vicente Ordonez, Joseph Redmon, and Ali Farhadi. "Xnor-net: Imagenet classification using binary convolutional neural networks." In European conference on computer vision, pp. 525-542. Springer, Cham, 2016.

[5] Liu, Zechun, Zhiqiang Shen, Marios Savvides, and Kwang-Ting Cheng. "Reactnet: Towards precise binary neural network with generalized activation functions." In European Conference on Computer Vision, pp. 143-159. Springer, Cham, 2020.

[6] Bethge, Joseph, Christian Bartz, Haojin Yang, Ying Chen, and Christoph Meinel. "MeliusNet: An Improved Network Architecture for Binary Neural Networks." In Proceedings of the IEEE/CVF Winter Conference on Applications of Computer Vision, pp. 1439-1448. 2021.

---

### Decision · Program_Chairs · 2022-01-20

**Decision:**

Reject

**Comment:**

### Description
The authors note that the most recent and successful in terms of accuracy binary networks are in fact combining binary and floating-point computations, in particular have residual full-precision paths, with their parameters, that connect all the way to the output. Although such paths are made lightweight, they can be a bottleneck with respect to the energy consumption, memory and latency. The paper proposes a novel binary neural network model that uses only binary operations (except in the first and last layers). It is proposed to estimate the energy efficiency of binary networks more accurately, using hardware design compilers.

### Decision
Reviewers came to a consensus that the proposed BNN architecture claimed in the paper to be the main novelty, while it is indeed quite distinct from the mainstream and best performing BNNs, does not propose novel solutions with respect to the total state of the art. This is our main reason for rejection. The paper makes a great effort in steering the development of BNNs towards more energy efficient models, by carefully estimating the potential energy consumption of different models, using specialized software for design and simulation of hardware needed to run particular models. This is proposed not as a practical solution for industry but rather as a way to measure the potential efficiency of different models. While this was recognized as a great effort, some questions remained regarding fairness of comparison and possibility to reproduce the results by non-experts in order that other developers could estimate and compare potential efficiency of their models. Additionally, it is unclear how power efficiency of a model with $k$ slices (BoolNet) differs from that of a $k$-bit quantized network.

### Details
First of all, I very much like the motivation for the work: to aim at a design of BNN that would be more efficient in terms of speed and energy and to measure that efficiency more precisely. I am not an expert in the hardware, however I do share the concern in this paper that full precision residual paths and blocks would incur a larger latency and higher amount of computation (more chip area, more energy,...). The fact that residual connections in BiRealNet make it necessary to read and write full precision feature maps to the global memory was non-obvious to me. The current state of the art reports binary and floating point operations and largely ignores the necessary memory operations, which as authors argue are the real performance bottleneck.

The main claimed contribution of the paper is the design of a novel model. This was the main point of concert. It is indeed innovative with respect to the current trend of the best performing binary networks to make a "pure" binary network. However it is hard to agree that removing some of the components from BiRealNet or models based on it and going back to simpler architectures which were used before can be a contribution. Specifically, plain non-residual BNNs were the very well-known pioneering works [r1, r2]. The fact that BN in front of binarization can be simplified is trivial and well known, e.g. [r3]. Using multiple binary activations through power-of-two coding or uniform thresholds has been considered multiple times [r2, r6, ABCNet]. Same for group-wise convolutions. I have not seen residual connections in the form of shuffle-net so far, but it can also be considered as a rather standard trick. Using stride instead of max pooling or average pooling is another well known trick. Many of these solutions are in fact used in recent works, e.g. [r4] has no skip-connections, combines BN with sign at the inference time and uses strided convolutions for downsampling. To summarize, the submission does not appear to propose novel modelling solutions relative to the total state of the art.
Furthermore, [r3] is closely related in that they investigated the energy efficiency (however looking at the energy consumption of individual operations only) and with a similar motivation developed binary networks without 32 bit residual connections. Please see also other references pointed out by reviewers.

### Unclear in the paper:
-What is the meaning of $F$ column in Fig4a for BaseNet / Boolnet?
-Why grouped convolution with input channels 256k and k groups has output of 256 channels in fig 3a.
If the outputs from each group are summed, isn't it equivalent to a full convolution 256k x 256?
-What do the authors mean by 3x3 depth-wise convolution?
- The local adaptive shifting module is discussed inside the paragraph describing MS-BConv. Is it a part of MS-BConv or not?
- It seems that with $k$ slices, there is $k$ times more bits per activation used and $k$ times more bits per weight used (because the channel width is multiplied by $k$).
It must be therefore equivalent in terms of the power consumption to a network that uses $k$ bits per activation and weight.
Using these $k$ bits to represent uniform slices rather than powers of two slices appears inferior in terms of quantization error and accuracy. Indeed, many works have successfully quantization different models down to $4$ bits without a loss in accuracy.

### Related work:

Rather than reviewing different methods for making networks more efficient, a deeper review of BNNs would be more helpful, in particular
looking at the works that are closer to the hardware, such as "in-memory computing", "neuromorphic computing", [r5, r6].

### Discussion

A well-justified way to measure the potential energy efficiency of BNNs would be an excellent contribution that could standardize comparison and drive the development of BNNs in the energy-efficient direction. Unfortunately,
it does not appear at the moment that non-experts in hardware and design compilers could repeat the compilation and simulation of accelerators as the authors proposed. A simplified estimation method is needed that can be used in python for any model composed of some standard blocks. The authors seem to be in a good position to propose and validate such a simplified estimator. To start with may I suggest to clarify the following questions:
- Do we need power for the operation pipelines for different operation types (cache, global memory) that are not currently used?
- Are the arithmetic operations implemented in hardware to optimize energy or the throughput?
- Can we assume that all latencies can be masked by parallelism?
- Is it a good approximation to assume that a convolution (with an efficient implementation and large enough cache) needs to read the input only once?
- Can any coordinate-wise transform be appended to the preceding transform on the fly, canceling a read-write in between?
- What is a reasonable estimate of a cost for float32 operations? It seems from the quantization literature that all such operations can be safely quantized to e.g. 8 bit representations without a loss of accuracy.

[r1] Courbariaux et al. 2016, Binarized Neural Networks: Training Neural Networks with Weights and Activations Constrained to +1 or −1]

[r2 Hubara 2018: Quantized Neural Networks: Training Neural Networks with Low Precision Weights and Activations, JMLR]

[r3 Ding et al. 2019: Regularizing Activation Distribution for Training Binarized Deep Networks]

[r4 Livochka et al. Initialization and Transfer Learning of Stochastic Binary Networks From Real-Valued Ones]

[r5 Baskin et al. Streaming Architecture for Large-Scale Quantized Neural Networks on an FPGA-Based Dataflow Platform]

[r6 Umuroglu et al. FINN: A Framework for Fast, Scalable Binarized Neural Network Inference]